# Impaired mRNA splicing and proteostasis in preadipocytes in obesity-related metabolic disease

**Julia Sánchez-Ceinos[1,2], Rocío Guzmán-Ruiz[1,2]\*, Oriol Alberto Rangel-Zúñiga[2,3], Jaime López-Alcalá[1,2], Elena Moreno-Caño[1], Mercedes Del Río-Moreno[2,4], Juan Luis Romero-Cabrera[3], Pablo Pérez-Martínez[3], Elsa Maymo-Masip[5,6], Joan Vendrell[5,6], Sonia Fernández-Veledo[5,6], José Manuel Fernández-Real[2,7], Jurga Laurencikiene[8], Mikael Rydén[8], Antonio Membrives[9], Raul M Luque[2,4], José López-Miranda[2,3], María M Malagón[1,2]\***

[1]Department of Cell Biology, Physiology and Immunology, Instituto Maimónides de Investigación Biomédica de Córdoba (IMIBIC)/University of Córdoba/Reina Sofía University Hospital, Córdoba, Spain; [2]CIBER Fisiopatología de la Obesidad y Nutrición (CIBERobn), Instituto de Salud Carlos III, Madrid, Spain; [3]Lipids and Atherosclerosis Unit, Department of Internal Medicine, IMIBIC, Córdoba, Spain; [4]OncObesity and Metabolism Group. Department of Cell Biology, Physiology and Immunology, IMIBIC/University of Córdoba/Reina Sofía University Hospital, Córdoba, Spain; [5]CIBER de Diabetes y Enfermedades Metabólicas Asociadas (CIBERdem), Instituto de Salud Carlos III, Madrid, Spain; [6]Hospital Universitari de Tarragona Joan XXIII, Institut d´Investigació Sanitària Pere Virgili Universitat Rovira i Virgil, Tarragona, Spain; [7]Department of Diabetes, Endocrinology and Nutrition, Dr. Josep Trueta University Hospital, and Eumetabolism and Health Group, Girona Biomedical Research Institute (IdibGi), Girona, Spain; [8]Lipid Laboratory. Department of Medicine Huddinge/Karolinska Institute (KI)/Karolinska University Hospital, Stockholm, Sweden; [9]Unidad de Gestión Clínica de Cirugía General y Digestivo, Sección de Obesidad, Reina Sofia University Hospital, Córdoba, Spain

\*For correspondence:
bc2gurur@uco.es (RG-R);
bc1mapom@uco.es (MMM)

**Competing interest:** The authors declare that no competing interests exist.

**Abstract** Preadipocytes are crucial for healthy adipose tissue expansion. Preadipocyte differentiation is altered in obese individuals, which has been proposed to contribute to obesity-associated metabolic disturbances. Here, we aimed at identifying the pathogenic processes underlying impaired adipocyte differentiation in obese individuals with insulin resistance (IR)/type 2 diabetes (T2D). We report that down-regulation of a key member of the major spliceosome, *PRFP8*/PRP8, as observed in IR/T2D preadipocytes from subcutaneous (SC) fat, prevented adipogenesis by altering both the expression and splicing patterns of adipogenic transcription factors and lipid droplet-related proteins, while adipocyte differentiation was restored upon recovery of *PRFP8*/PRP8 normal levels. Adipocyte differentiation was also compromised under conditions of endoplasmic reticulum (ER)-associated protein degradation (ERAD) hyperactivation, as occurs in SC and omental (OM) preadipocytes in IR/T2D obesity. Thus, targeting mRNA splicing and ER proteostasis in preadipocytes could improve adipose tissue function and thus contribute to metabolic health in obese individuals.

## Introduction

Adipose precursor cells, the preadipocytes, are essential for the maintenance of adipose tissue homeostasis, regeneration, and expansion (*Berry et al., 2016*). Preadipocytes differentiation into adipocytes, i.e. adipogenesis, enables adipocyte turnover and adipose tissue growth and ensures adipose tissue plasticity to accommodate surplus energy (*Arner et al., 2010*; *Sarjeant and Stephens, 2019*; *Spalding et al., 2008*). It has been proposed that the inability for recruiting new adipose cells together with the functional impairment of hypertrophied adipocytes that occur in obesity contribute to lipid spill over from the adipose tissue (*Lessard and Tchernof, 2012*). Ectopic fat deposition and lipotoxicity in non-adipose organs have been associated with the development of insulin resistance (IR) and type 2 diabetes (T2D) (*Engin and Engin, 2017*). Thus, increasing adipogenesis appears as a valuable strategy to facilitate healthy adipose tissue expansion and ensure metabolic health (*Ghaben and Scherer, 2019*).

Adipocyte differentiation involves striking changes in size, metabolism, and responsiveness to signals. This process, in turn, relies on major changes in gene expression programs regulating mRNA and protein production (*Mota de Sá et al., 2017*). An increasing body of evidence shows that mRNA processing and, in particular, alternative splicing, is crucial for genome reprogramming during cell differentiation (*Fiszbein and Kornblihtt, 2017*). In this line, the recent identification of PPARγ protein variants with opposing effects on preadipocyte differentiation, supports a role for alternative splicing in adipogenesis (*Aprile et al., 2020*; *Aprile et al., 2018*; *Aprile et al., 2014*). However, the splicing components relevant to adipogenesis and the cellular events regulated by alternative splicing during adipocyte differentiation have been scarcely explored (*Lin, 2015*), and it is yet to be established whether alternative splicing is modified in human obesity.

Another crucial mechanism preserving precursor cell function relates to protein homeostasis (i.e. proteostasis), which maintains the capacity of cells to expand in order to sustain tissue growth and regeneration (*Noormohammadi et al., 2018*). Several lines of evidence support an important role for the endoplasmic reticulum (ER) protein quality control system in the regulation of adipogenesis (*Lowe et al., 2011*). In fact, the unfolded protein response (UPR) is perturbed in the obese adipose tissue and has been proposed to contribute to the pathology of obesity (*Yilmaz, 2017*). By contrast, it is still unknown whether the other component of the protein control system, the ER-associated protein degradation (ERAD), which is crucial for protecting cells against accumulation of misfolded/unfolded proteins and proteotoxicity (*Hwang and Qi, 2018*), is altered in the obese adipose tissue. Importantly, genetic ablation of the core ERAD protein, SEL1L, in mouse adipocytes causes lipodystrophy and postprandial hyperlipidaemia (*Sha et al., 2014*).

Here, we set out an iTRAQ-LC-MS/MS proteomic approach for the analysis of subcutaneous (SC) and omental (OM) preadipocytes from obese individuals with normoglycaemia (NG) and T2D in order to identify altered molecular pathways that may contribute to metabolic disease in obesity. Down-regulation of multiple components of the splicing machinery was observed in SC preadipocytes from obese individuals with insulin resistance (IR) or T2D, as compared to NG obesity. This, together with the observation that adipogenesis can be modulated by regulating the expression levels of a key spliceosome component, *PRFP8*/PRP8, support a role for alternative splicing in the development of obesity-associated metabolic complications. In addition, our studies show that not only the UPR but also the ERAD system are altered in human SC and OM preadipocytes from IR/T2D obese subjects, and that this condition (i.e., hyperactivated ERAD), when mimicked in vitro, prevented adipogenesis. Our results provide novel mechanistic explanations for the impaired adipogenic capacity observed in IR/T2D obesity that relates to both mRNA and ER-proteostasis disturbance.

## Results

### Preadipocytes display distinct features in obesity-related IR/T2D

Preadipocytes from SC and OM adipose tissue were isolated from 78 morbidly obese subjects (BMI <40 kg/m$^2$) (hereinafter referred to as obese individuals) (*Table 1*), who were subclassified into three groups [normoglycemic (NG Obese): Glucose <100 mg/dL, HbA1c < 5.7 %; impaired fasting glucose (IFG Obese): Glucose 100–126 mg/dL, HbA1c 5.7–6.5%; and with type 2 diabetes (T2D Obese): Glucose >126 mg/dL, HbA1c > 6.5%], according to the criteria of the American Diabetes Association (*ADA, 2021*). IFG individuals exhibited significantly higher HOMA-IR values than NG

**Table 1.** Anthropometric and biochemical characteristics of study subjects from cohort 1.

| | NG obese | IR obese | T2D obese |
|---|---|---|---|
| N | 30 | 30 | 18 |
| Gender (female/male) | 15 / 15 | 15 / 15 | 11 / 7 |
| Post-menopause (n, %) | 2 (13) | 2 (13) | 3 (27) |
| Lipid-lowering therapy (n, %) | 0 (0) | 5 (17) | 4 (22) |
| Antidiabetic therapy (n, %) | 0 (0) | 3 (10) | 8 (44) |
| Antihypertensive therapy (n, %) | 1 (3) | 6 (20) | 5 (28) |
| Age (years) | 43 ± 2 | 44 ± 2 | 46 ± 2 |
| Weight (kg) | 140.4 ± 7.0 | 153.4 ± 10.0 | 145.4 ± 7.5 |
| Height (m) | 1.67 ± 0.03 | 1.69 ± 0.02 | 1.65 ± 0.03 |
| Body mass index (kg/m$^2$) | 50.2 ± 2.1 | 52.8 ± 2.9 | 52.9 ± 1.9 |
| Fat mass (%) | 43.1 ± 1.9 | 41.2 ± 2.0 | 42.1 ± 1.7 |
| Lean mass (%) | 39.1 ± 3.2 | 36.1 ± 1.1 | 35.0 ± 1.0 |
| Water mass (%) | 23.0 ± 3.5 | 22.7 ± 1.6 | 22.9 ± 1.3 |
| Waist circumference (cm) | 144.2 ± 5.9 | 156.1 ± 8.1 | 149.9 ± 4.6 |
| Systolic pressure (mm/Hg) | 128.1 ± 2.3 | 127.0 ± 3.5 | 122.1 ± 2.6 |
| Diastolic pressure (mm/Hg) | 78.5 ± 4.4 | 75.5 ± 3.1 | 71.5 ± 2.6 |
| Fasting glucose (mg/dL) | 89.1 ± 1.7 | 105.4 ± 2.0 [aaa] | 157.3 ± 9.6 [aaa, bbb] |
| Fasting glucose (mmol/L) | 4.95 ± 0.10 | 5.85 ± 0.11 [aaa] | 8.73 ± 0.54 [aaa, bbb] |
| Fasting insulin (mU/L) | 15.4 ± 1.8 | 25.9 ± 2.9 [aa] | 17.9 ± 2.7 [b] |
| HbA1c (%) | 5.44 ± 0.06 | 6.22 ± 0.10 [aaa] | 8.43 ± 0.53 [aaa, bb] |
| HbA1c (mmol/mol) | 32.2 ± 0.6 | 40.9 ± 1.2 [aaa] | 64.8 ± 5.7 [aaa, bb] |
| HOMA-IR (units) | 3.42 ± 0.44 | 6.77 ± 0.74 [aa] | 6.74 ± 1.05 [aa] |
| Total cholesterol (mg/dL) | 167.4 ± 10.0 | 181.6 ± 8.3 | 198.1 ± 8.6 |
| LDL cholesterol (mg/dL) | 122.9 ± 11.1 | 112.9 ± 6.6 | 124.2 ± 8.5 |
| HDL cholesterol (mg/dL) | 39.5 ± 3.3 | 36.3 ± 2.1 | 36.7 ± 1.6 |
| Triglycerides (mg/dL) | 108.2 ± 7.0 | 132.7 ± 12.2 | 152.0 ± 12.9 [a] |
| Free fatty acids (mmol/L) | 66.7 ± 6.3 | 74.4 ± 7.2 | 83.5 ± 6.2 |
| C-reactive protein (mg/L) | 9.61 ± 1.68 | 11.4 ± 2.3 | 17.4 ± 5.3 |
| Uric acid (mg/dL) | 6.30 ± 0.32 | 7.03 ± 0.36 | 6.85 ± 0.68 |

NG = normoglycemic. **IR** = insulin-resistant. **T2D** = type 2 diabetes. **LDL** = low-density lipoprotein. **HDL** = high-density lipoprotein. **HbA1c** = glycated hemoglobin. **HOMA-IR** = homeostasis model assessment of insulin resistance. [a]P <0.05, [aa]P <0.01, [aaa]P <0.001 *vs.* NG Obese; [b]P <0.05, [bb]P <0.01, [bbb]P <0.001 *vs.* IR Obese. One-way ANOVA with Tukey's multiple comparisons test or Kruskal-Wallis with Dunn's multiple comparisons test (for parametric or non-parametric data, respectively) were used. Normality distribution was determined by Shapiro-Wilk normality test.

individuals and they will be referred to as insulin-resistant (IR) subjects (*Díaz-Ruiz et al., 2015*; *Tam et al., 2012*). No significant differences were found among groups in blood pressure, inflammation, or lipid parameters except triglycerides, whose levels were enhanced in T2D *vs.* NG obese groups.

Paired SC and OM adipose tissue samples were processed for the isolation of the stroma-vascular fraction (SVF) and mature adipocytes. Microscopic analysis of freshly isolated mature adipocytes revealed that these cells were larger in IR/T2D than in NG obese individuals (*Figure 1—figure supplement 1A*).

SC and OM preadipocytes were obtained after serial passaging (2–3 passages) of the corresponding SVFs following established methods (*Bunnell et al., 2008*; *Palumbo et al., 2018*; *Serena et al., 2016*; *Zhu et al., 2013*). The purity of the preadipocyte cultures was confirmed by the presence of the preadipocyte marker DLK1/PREF1 and the absence of the blood cell markers CD45 and CD14 (*Figure 1—figure supplement 1B*). Expression levels of the proliferation marker, *KI67*, and the proliferation rate, as assessed by cell count, were higher in SC than in OM preadipocytes, and both parameters decreased in IR/T2D (*Figure 1—figure supplement 1C and D*). Preadipocyte proliferation rate negatively correlated with obesity and IR markers (BMI, glucose, insulin, and HbA1c) (*Supplementary file 1*).

Microscopic observation of lipid droplets (LDs) showed that both SC and OM preadipocytes from NG obese subjects differentiated well, while those from IR/T2D obese groups differentiated poorly (*Figure 1—figure supplement 1E*). In this line, the mRNA levels of the adipocyte markers, *PPARG*, *FABP4,* and *ADIPOQ* were lower in IR obese subjects than in their NG counterparts throughout differentiation (*Figure 1—figure supplement 1F*).

## Comparative proteomics of human obese preadipocytes

In order to identify marker pathogenic pathways of dysfunctional preadipocytes in obesity-associated metabolic disease, we employed iTRAQ proteomics of SC and OM preadipocytes from obese individuals with extreme metabolic phenotypes, NG *vs.* T2D. This study enabled the identification of a total of 1758 proteins that were present in both SC and OM fat from NG and T2D obese individuals, thus defining the human obese preadipocyte proteome. According to GO Biological Process annotation, 55.5 % of these proteins were related to cellular and metabolic processes (*Figure 1—figure supplement 2A*), including metabolism of mRNA, proteins, glucose, and lipids (*Figure 1—figure supplement 2B*). All the proteomic data related to this study are available at the ProteomeXchange Consortium via the Proteomics IDEntifications (PRIDE) partner repository (*Perez-Riverol et al., 2019*) with the dataset identifier PXD015621.

Comparison of proteomic data from SC and OM preadipocytes revealed marked differences between the two types of cells regarding the number of differentially expressed proteins and protein-related functional pathways (*Figure 1—figure supplement 3*, and *Figure 1—source data 1*). To investigate enrichment for functional pathways in each depot, canonical pathway analysis of the proteomic data was performed using IPA. Among the top ten IPA canonical pathways that were overrepresented in SC preadipocytes in T2D obesity (*Figure 1—source data 2*), the highest ranked down-regulated pathway was the spliceosomal cycle (*Figure 1B*). The unfolded protein response (UPR) was also modified in SC preadipocytes from T2D obese group (*Figure 1B*), while pathways related to ER stress were up-regulated in OM preadipocytes (*Figure 1C*) in T2D *vs.* NG obesity.

Based on these findings, we next characterized the splicing machinery and ER control system in preadipocytes and their contribution to adipocyte differentiation.

## Splicing dysregulation as a marker of SC preadipocytes in obesity-related metabolic disease

To complement our proteomic observations, we assessed the expression of splicing-related genes in human preadipocytes using a microfluidic-based dynamic qPCR array comprising 45 splicing-related components (*Gahete et al., 2018*). Preadipocytes isolated from the SC and OM fat pads of NG, IR, and T2D obese individuals were employed for this study. Analysis of gene expression profiles by hierarchical clustering revealed a marked gene expression shift between NG and IR/T2D SC preadipocytes, which was not evident for OM preadipocytes (*Figure 2A*). SC preadipocytes from IR/T2D obese individuals exhibited lower expression levels of splicing-related genes than NG obese subjects, including components of major spliceosome (*PRPF8*, *RNU5*, *SF3B1 tv1*, *TCERG1*, and *U2AF1*), and splicing factors (*CELF1*, *MAGOH*, *RBM3*, *RBM45*, *SFPQ*, and *SNW1*) (*Figure 2A and B*, and *Figure 2—figure supplement 1*). Notably, only the minor spliceosome component, *RNU12*, was up-regulated in T2D *vs.* NG SC preadipocytes (*Figure 2A* and *Figure 2—figure supplement 1B*). No differences in OM preadipocytes were observed among the three groups of obese individuals (*Figure 2* and *Figure 2—figure supplement 1B*). These observations were largely confirmed in a second independent subset of samples obtained from additional obese individuals using RT-PCR (*Figure 2—figure supplement 2*).

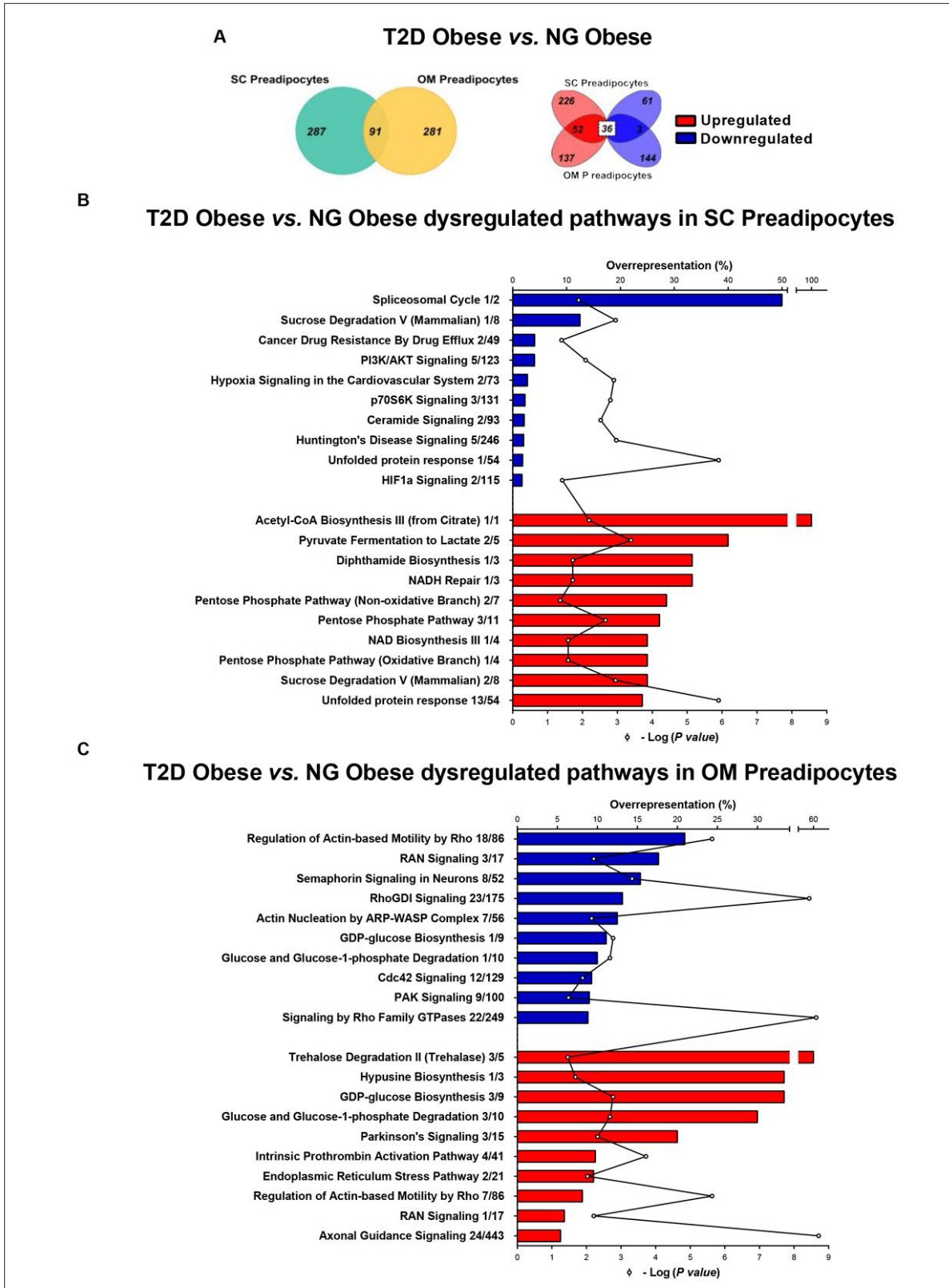

**Figure 1.** Comparative proteomic analysis of subcutaneous (SC) and omental (OM) preadipocytes from obese subjects with type 2 diabetes (T2D) *vs.* normoglycaemia (NG). Data correspond to individuals from cohort 1. (**A**) Venn diagrams showing overlap of differentially regulated proteins between T2D *vs.* NG obese subjects in SC and OM preadipocytes (left panel). Up-regulated and down-regulated proteins are indicated in the right panel. (**B**) Top10 over-represented canonical pathways in T2D *vs.* NG obese subjects in SC preadipocytes and (**C**) in OM preadipocytes according to Ingenuity

*Figure 1 continued on next page*

*Figure 1 continued*

Pathway Analysis (IPA). Blue and red bars indicate down-regulated and up-regulated pathways, respectively, in T2D *vs*. NG obese subjects. Numbers indicate the number of identified proteins/total proteins annotated to the pathway. Black line indicates -Log$_2$(p *value*). Samples from wo to three individuals per group and fat depot were pooled and used for two separate iTRAQ experiments ( n = 5–6 subjects/group/fat depot). Data normality was tested by Shapiro-Wilk test and Student's t test was used, a ± 1.5 fold change with p < 0.05 was set as the threshold for categorizing up- and down-regulated proteins. Canonical pathway analysis was performed using IPA (see Materials and ethods section). The online version of this article includes the following figure supplements for *Figure 1*, *Figure 1—figure supplement 1*, *Figure 1—figure supplement 2*, and *Figure 1—figure supplement 3*; and the following source data for *Figure 1—source data 1* and *Figure 1—source data 2*.

The online version of this article includes the following figure supplement(s) for figure 1:

**Source data 1.** Common pathways of normoglycemic (NG) and with type 2 diabetes (T2D) morbidly obese subjects (cohort 1) in the subcutaneous (SC) vs omental (OM) preadipocytes proteome according to Ingenuity Pathway Analysis (IPA).

**Source data 2.** Fat depot-specific canonical pathways in the type 2 diabetes (T2D) *vs*. normoglycemic (NG) morbidly obese subjects (cohort 1) proteome according to Ingenuity Pathway Analysis (IPA).

**Figure supplement 1.** Characterization of SC and OM adipocytes and preadipocytes from NG, IR, and T2D morbidly obese individuals (cohort 1).

**Figure supplement 2.** Characterization of the human preadipocyte proteome.

**Figure supplement 3.** Comparative proteomic analysis of subcutaneous (SC) *vs*. omental (OM) preadipocytes from NG and T2D morbidly obese subjects (cohort 1).

The splicing machinery was also investigated in preadipocytes from a second cohort (*Supplementary file 2*), comprising lean subjects (BMI <25 kg/m$^2$) and NG and T2D individuals with simple obesity (BMI 30–35 kg/m$^2$) (hereinafter referred to as simple obesity) (*Figure 2C* and *Figure 2—figure supplement 3*). These results revealed significantly higher expression levels of main components of the major spliceosome (*PRPF8* and *SF3B1 tv1*) in SC preadipocytes from NG obese *vs*. lean individuals (*Figure 2C* and *Figure 2—figure supplement 3A*). A trend to increase was also observed for the minor spliceosome component, *RNU12*, and the splicing factors, *CELF1* and *SNW1*, although this did not reach statistical significance (*Figure 2—figure supplement 3B and C*). Notably, and similar to that observed for morbidly obese individuals from cohort 1 (*Figure 2A* and *Figure 2—figure supplements 1 and 2*), a significant decrease in the expression levels of all the genes tested, except *RNU12*, was observed in T2D *vs*. NG individuals with simple obesity (*Figure 2C* and *Figure 2—figure supplement 3*). No differences were observed in OM preadipocytes when groups of cohort two were compared (*Figure 2C* and *Figure 2—figure supplement 3*).

### *PRPF8*/PRP8 expression studies in human preadipocytes

One of the most highly expressed genes in SC preadipocytes from NG obese individuals that was significantly down-regulated in both IR and T2D obese subjects as compared to NG SC preadipocytes, was *PRPF8*/PRP8. Specifically, mRNA and protein levels of this key component of the major spliceosome were reduced by 51% and 56%, respectively, in IR SC preadipocytes, and by 49% and 82% in T2D SC preadipocytes as compared to NG levels (*Figure 2A and B*). No differences in *PRPF8*/PRP8 expression were observed among groups in OM preadipocytes (*Figure 2A and B*). Differentiation studies revealed that while *PRPF8* expression levels remained low throughout adipogenesis in IR preadipocytes, *PRPF8* levels peaked at day 3 (D3) of differentiation and thereafter remained above D0 levels in NG preadipocytes (*Figure 2D*). *PRPF8* mRNA remained constant or decreased at the end of differentiation of NG and IR OM preadipocytes, respectively (*Figure 2D*).

### *PRPF8*/PRP8 down-regulation in SGBS preadipocytes impairs adipogenesis

We employed a siRNA strategy to down-regulate *PRPF8* gene expression levels in preadipocytes to mimic the conditions found in IR/T2D obese SC preadipocytes as compared to NG obesity. These studies were carried out using the human SC adipocyte cell line, SGBS cells. As observed for NG SC human primary preadipocytes, *PRPF8* mRNA levels reached a peak at early stages of SGBS cell differentiation (D4) (*Figure 3A*). siRNA treatment of SGBS preadipocytes (D4) decreased by 67% and 65% *PRPF8* mRNA and protein levels, respectively, at day 3 post-transfection (D7), without changing cell viability (*Figure 3—figure supplement 1A-C*). Morphometric evaluation of Oil-Red O staining in confocal micrographs revealed that *PRPF8*-silenced preadipocytes accumulated more but smaller

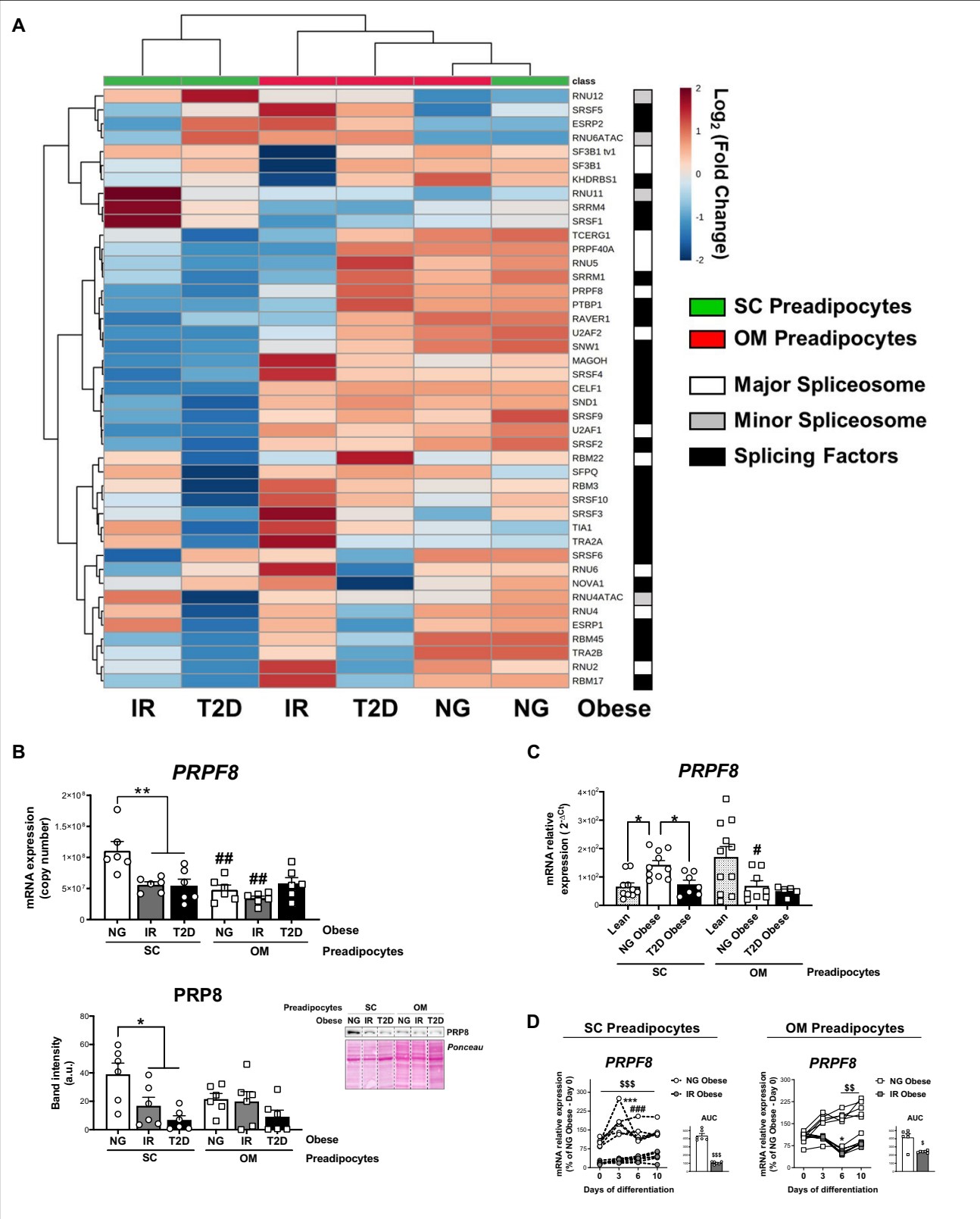

**Figure 2.** Downregulation of the splicing machinery in subcutaneous (SC) preadipocytes is associated with obesity-related insulin resistance (IR) and type 2 diabetes (T2D). (**A**) Hierarchical clustering dendrogram heatmap analysis of splicing-related genes in SC (green) and OM (red) preadipocytes from normoglycemic (NG), IR, and T2D morbidly obese subjects (cohort 1) ( n = 6, 1 technical replicate each) measured by qPCR dynamic array. Rows stand for splicing-related genes (white, mayor spliceosome; grey, minor spliceosome; black, splicing factors), while columns stand for subject groups. The

*Figure 2 continued on next page*

*Figure 2 continued*

scale in the colour bar represents -Log$_2$(Fold Change). (**B**) mRNA levels of *PRPF8* measured by qPCR dynamic array (upper graph) and representative blot and protein level quantification of PRP8 (lower graph) in SC and OM preadipocytes from NG, IR, and T2D morbidly obese subjects (cohort 1; n = 6, 1 technical replicate each) *p < 0.05, **p < 0.01 *vs.* NG and/or IR subjects, ##p < 0.01 *vs.* SC preadipocytes from the same subjects. (**C**) mRNA levels of *PRPF8* in SC and OM preadipocytes from lean, and NG and T2D subjects with simple obesity (cohort 2; n = 5–11, 1 technical replicate each). *p < 0.05 *vs.* lean and/or NG obese subjects, #p < 0.05 *vs.* SC preadipocytes from the same subjects. (**D**) *PRPF8* mRNA levels and area under the curve (AUC) during in vitro differentiation of SC and OM preadipocytes from NG and IR morbidly obese subjects (cohort 1; n = 6, 1 technical replicate each). *p < 0.05, ***p < 0.001 *vs.* *PRPF8* mRNA levels at Day 0; ###p < 0.001 *vs.* *PRPF8* mRNA levels at Day 3; $< 0.05, $$p < 0.01, $$$p < 0.001 *vs.* NG. Data are presented as mean ± standard error of the mean (S.E.M.). One-way ANOVA with Tukey's multiple comparisons test or Kruskal-Wallis with Dunn's multiple comparisons test (for parametric or non-parametric data, respectively) were used for B and C; two-way ANOVA was used for D. Normality distribution was determined by Shapiro-Wilk normality test. The online version of this article includes the following figure supplements for *Figure 2—figure supplement 1*, *Figure 2—figure supplement 2*, and *Figure 2—figure supplement 3*.

The online version of this article includes the following figure supplement(s) for figure 2:

**Figure supplement 1.** mRNA levels of components of the major spliceosome (**A**), minor spliceosome (**B**), and splicing factors (**C**) in SC and OM preadipocytes from NG, IR, and T2D morbidly obese subjects (cohort1; n = 6, 1 technical replicate each) measured by qPCR dynamic array.

**Figure supplement 2.** mRNA levels of components of the major spliceosome (**A**), minor spliceosome (**B**), and splicing factors (**C**) in SC and OM preadipocytes from NG, IR, and T2D morbidly obese subjects (cohort 1; n = 6, 1 technical replicate each) measured by RT-PCR.

**Figure supplement 3.** mRNA levels of representative components of the splicing machinery, including the major spliceosome (*SF3B1 tv1*) (**A**), minor spliceosome (*RNU12*) (**B**), and splicing factors (*CELF1* and *SNW1*) (**C**) in SC and OM preadipocytes from lean individuals and NG and T2D subjects with simple obesity (cohort 2; n = 5–11, 1 technical replicate each).

LDs than control cells, which resulted in an increase in the total lipid content in cells exposed to *PRPF8* siRNA (*Figure 3C*). *In silico* analysis of CLIP_Seq data using ENCORI (The Encyclopedia of RNA Interactomes) (*Li et al., 2014*) revealed both adipogenic (*PPARG* and *SREBF1*) and LD biogenesis and growth markers (*BSCL2*, *CIDEB*, and *CIDEC*) as PRP8 target genes (*Figure 3—source data 1*). Enrichment analysis of PRP8-RNA interactions revealed pathways such as insulin signalling, adipokine signalling, and fatty acid metabolism to be significantly overrepresented (*Figure 3—source data 1*). Similarly, the HumanBase tool (*Greene et al., 2015*) identified several adipogenesis-related genes, including *ADIPOQ*, *CAV1*, and *CD36*, among the Top10 genes showing functional interactions with *PRPF8* (*Figure 3—source data 2*). In accordance with the *in silico* data, expression of total *PPARG* and *SREBF1* transcript contents and the abundances of some of their isoforms were altered in *PRPF8*-silenced cells (*Figure 3D and E*). In particular, mRNA and protein levels of the fat-specific *PPARG* isoform, *PPARG-2*, were up-regulated upon *PRPF8* silencing (*Figure 3D and E*). Decreased levels and/or altered splicing patterns of *BSCL2*, *CIDEB*, and *CIDEC*, were also observed in silenced cells (*Figure 3F*). Other genes showing significant changes in *PRPF8*-silenced cells included enzymes involved in lipid synthesis (FAS and DGAT2), and the LD-associated protein, ADRP, while neither PLIN1 nor HSL/pHSL were modified (*Figure 3G*). *PRPF8* silencing also activated stress responses in SGBS adipocytes, as indicated by the increased protein levels of the ER stress marker, CHOP, the immuno-proteasome compkeonent, PSMB8, and CASP3/Pro-CASP3 ratio (*Figure 3G*). Contrarily to control cells, *PRPF8*-silenced cells did not respond to an insulin challenge (*Figure 3H*). Most of the effects depicted for silenced SGBS cells at D7 remained significant at D10 (*Figure 3—figure supplement 1*). Nevertheless, silenced cells at D10 still exhibited abundant LDs but of small size, thus resulting in a drastic decrease in the total amount of lipids stored in *PRPF8* siRNA-treated cells as compared with controls (*Figure 3I*). *PPARG-2* mRNA and protein, and the late adipocyte markers, *ADIPOQ* and *FABP4*, were down-regulated at D10 in *PRPF8*-silenced cells (*Figure 3—figure supplement 1D, E and H* ).

Notably, recovery of PRP8 protein levels by co-transfection of *PRPF8*-silenced SGBS cells with an expression vector coding for this protein (*PRPF8*-pcDNA3.1) reverted the effects induced by *PRPF8* silencing on adipocyte and LD markers, both at D7 (data not shown) and D10 (*Figure 3—figure supplement 1*). Silencing-induced changes in both LD number and size were also reverted in rescue experiments by *PRPF8* re-expression (*Figure 3I*). Silencing experiments using human adipose-derived stem cells (hADSCs) gave similar results to those observed in SGBS cells (*Figure 3—figure supplement 2*).

We also examined the contribution of other splicing genes to adipocyte differentiation, namely the SF3B complex and the splicing factor, SFPQ. Specifically, since SC preadipocytes from IR/T2D obese

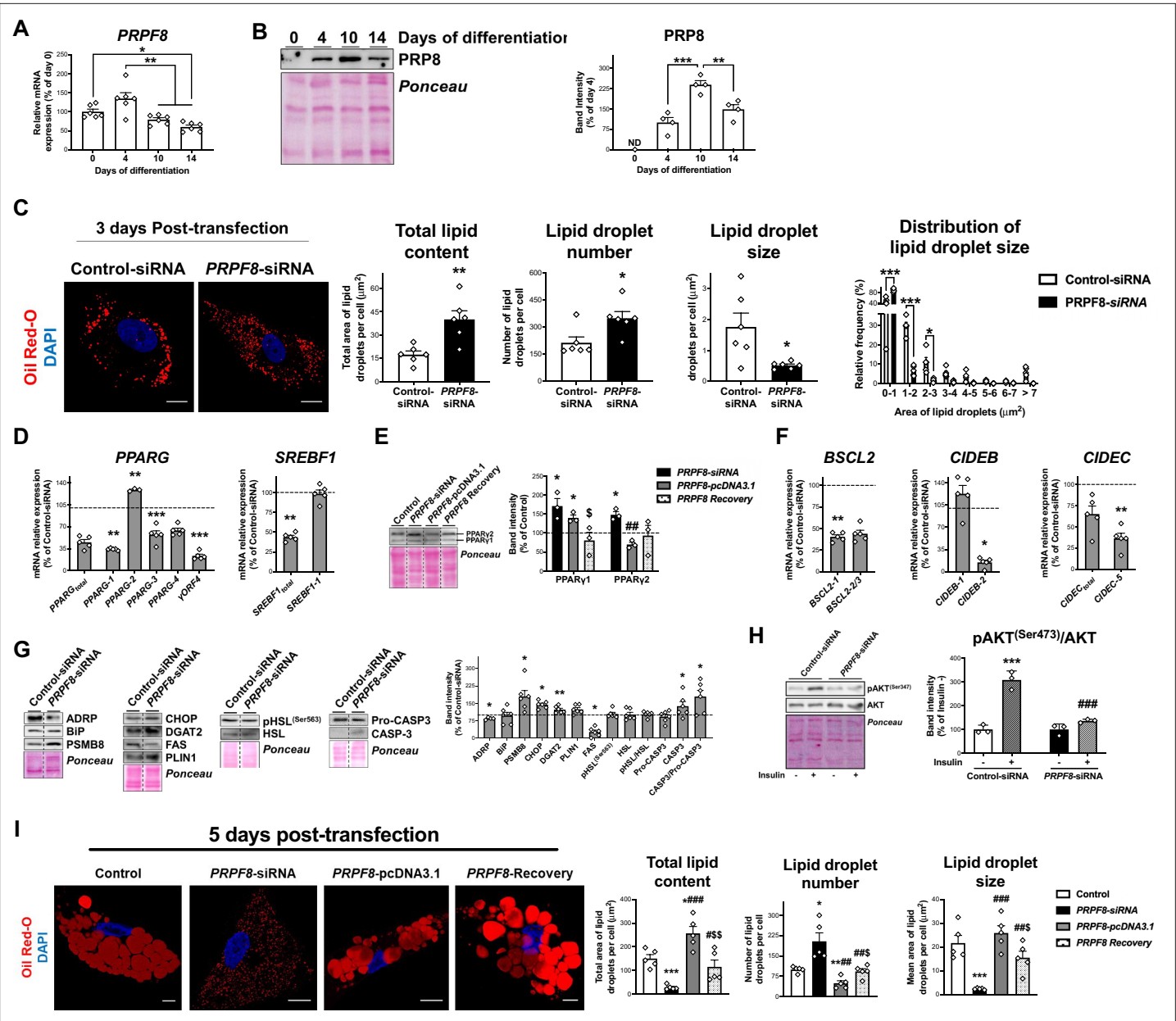

**Figure 3.** *PRPF8* silencing impairs adipogenesis in SGBS cells. (**A**) *PRPF8* mRNA levels in SGBS cells during adipogenesis. (**B**) Representative blot and protein quantification of PRP8 content in SGBS cells during adipogenesis. (4–6 replicate studies, 3 technical replicates each). (**ND**), non-detected. *p < 0.05, **p < 0.01, ***p < 0.001 vs. preceding differentiation days. (**C**) Representative confocal micrographs of SGBS adipocytes 3 days post-transfection (differentiation day 7) with control or *PRPF8*-siRNA stained with Oil Red-O [lipid droplets (LDs), red] and DAPI (nucleus, blue). Morphometric analysis of LDs was carried out using ImageJ software. Scale bar = 10 μm. (six replicate studies, 10 cells each). mRNA levels of splicing variants of the transcription factors, *PPARG* and *SREBF1* (**D**), and representative blot and protein level quantification of PPARγ1 and PPARγ2 (**E**), and of the LD-related proteins, *BSCL2*, *CIDEB*, and *CIDEC* (**F**), in SGBS adipocytes 3 days post-transfection. (**G**) Representative blots and protein level quantification of ADRP, BiP, PSMB8, CHOP, DGAT2, FAS, PLIN1, pHSL(Ser563), HSL, Pro-caspase3, and Caspase-3 in SGBS adipocytes 3 days post-transfection. (3–6 replicate studies, 3 technical replicates each). *p < 0.05, **p < 0.01, ***p < 0.001 vs. control; ##p < 0.01 vs. *PRPF8*-siRNA; $p < 0.05 vs. *PRPF8*-pcDNA3.1. (**H**) Representative blots and protein level quantification of pAKT(Ser473) and AKT in SGBS adipocytes 3 days post-transfection treated with/without insulin (100 nmol/L, 15 min). (three replicate studies, 3 technical replicates each). ***p < 0.001 vs. Control-siRNA -Insulin; ###p < 0.001 vs. control-siRNA+ Insulin. (**I**) Representative confocal micrographs of SGBS adipocytes 5 days post-transfection (day 10 of differentiation) with control, *PRPF8*-siRNA or *PRPF8*-pcDNA3.1 alone, or in combination (*PRPF8* Recovery) stained with Oil Red-O (LDs, red) and DAPI (nucleus, blue). Morphometric analysis of LDs was carried out using ImageJ software. Scale bar = 10 μm. (five replicate studies, 10 cells each). *p < 0.05, **p < 0.01, ***p < 0.001 vs. control; #p < 0.05, ##p < 0.01, ###p < 0.001 vs. *PRPF8*-siRNA; $p < 0.05. $$p < 0.01 vs. *PRPF8*-pcDNA3.1. Data are presented as mean ± standard error of the mean (S.E.M.). One-way ANOVA with Tukey's multiple comparisons test or Kruskal-Wallis with Dunn's multiple comparisons test (for parametric or non-parametric data,

*Figure 3 continued on next page*

*Figure 3 continued*

respectively) were used for A and B; unpaired t test or Mann Whitney test (for parametric or non-parametric data, respectively) were used for C-G and I; and two-way ANOVA was used for H. Normality distribution was determined by Shapiro-Wilk normality test. The online version of this article includes the following figure supplements for *Figure 3—figure supplement 1*, *Figure 3—figure supplement 2*, and *Figure 3—figure supplement 3*, and the following source data for *Figure 3—source data 1* and *Figure 3—source data 2*.

The online version of this article includes the following figure supplement(s) for figure 3:

**Source data 1.** Analysis of CLIP_Seq data (target binding sites and related pathways) from *PRPF8*-silenced HepG2 and K562 cells provided by The Encyclopedia of RNA Interactomes (ENCORI) database.

**Source data 2.** Adipose tissue-specific network of the functional interactions of *PRPF8*, *SF3B1*, and *SFPQ* according to HumanBase tool database, and binding sites of SFPQ according to SpliceAid-F.

**Figure supplement 1.** mRNA levels of *PRPF8* in SGBS adipocytes at day 3 and at day 5 post-transfection (D7 and D10 of differentiation, respectively) (three replicate studies, 4 technical replicates each) (**A**), and representative blot and protein level quantification of PRP8 in SGBS adipocytes at day 5 post-transfection (three replicate studies, 3 technical replicates each) (**B**) with control, *PRPF8*-siRNA or *PRPF8*-pcDNA3.1, alone or in combination (*PRPF8* Recovery).

**Figure supplement 2.** Effects of *PRPF8* silencing in differentiating human adipose-derived stem cells (hADSCs).

**Figure supplement 3.** Contribution of the spliceosome components, SF3B1 and SFPQ, to SGBS adipocyte differentiation.

individuals showed diminished *SF3B1 tv1* expression levels (*Figure 3—figure supplements 1 and 2*), we exposed SGBS cells to pladienolide-B, which binds to the SF3B complex and inhibits pre-mRNA splicing via targeting splicing factor SF3B1 (*Aouida et al., 2016*; *Cretu et al., 2018*). Exposure to pladienolide-B impaired lipid accumulation in SGBS cells (*Figure 3—figure supplement 3A*). This concurred with, among other changes, diminished expression levels of *PPARG-1*, *PPARG-2*, and both total *SREBF1* and *SREBF1-1*, while *PPARG-4* and *BSCL2-2/3* expression increased (*Figure 3—figure supplement 3B and C*). On the other hand, *SFPQ* down-regulation by siRNA treatment of SGBS cells caused significant changes in the expression of only a few genes, i.e., up-regulation of total *SREBF1* and *SREBF1-1* and down-regulation of *PPARG-4* (*Figure 3—figure supplement 3D-G*). Nevertheless, silenced SGBS cells exhibited reduced lipid content, mostly due to the increase in the number of LDs (2.4-fold), while LD size decreased as compared with control cells (2.7-fold) (*Figure 3—figure supplement 3E*).

## Analysis of *PRPF8*/PRP8 effects on human primary preadipocytes

Based on our findings in SGBS preadipocytes, we carried out targeted silencing and overexpression studies of *PRPF8*/PRP8 in primary preadipocytes obtained from SC adipose tissue samples of NG, IR, and T2D obese individuals (*Figure 4*).

First, we examined the effect of *PRPF8* silencing in SC preadipocytes from obese NG patients using a specific siRNA as a mean to mimic the down-regulation of this gene found in SC preadipocytes from obese subjects with IR/T2D (*Figure 2A and B*, *Figure 2—figure supplement 1* and *2*). As shown in *Figure 4B*, SC preadipocytes from NG obese individuals exhibited lower lipid content and changes in LD size and number when silenced for *PRPF8*. In addition, *PRPF8* down-regulation caused numerical decreases, that in most cases reached statistical significance, in the expression levels of both total *PPARG* and *SREBF1* and their isoforms as compared to mock-transfected SC preadipocytes from NG obese subjects (*Figure 4C–D*). Notably, rescue experiments by transfection of silenced NG obese preadipocytes with the *PRPF8*-pcDNA3.1 expression vector showed restored expression of most of the genes tested, which was accompanied by a recovery of LD content (*Figure 4*).

Second, in order to test whether recovery of *PRPF8*/PRP8 levels could improve the differentiation capacity of IR and T2D obese SC preadipocytes, overexpression studies using the *PRPF8*-pcDNA3.1 expression vector were carried out (*Figure 4*). After transfection, both IR and T2D obese SC preadipocytes exhibited *PRPF8* expression levels comparable to those of SC preadipocytes from obese NG patients (*Figure 4A*). *PRPF8* expression recovery upon *PRPF8*-pcDNA3.1 transfection increased the expression of several *PPARG* isoforms (*PPARG-2*, *PPARG-3*, *PPARG-4*, and *γORF4*) to control levels (i.e. SC preadipocytes in NG obesity), especially in IR preadipocytes (*Figure 4C*). Similar results were observed for both total *SREBF1* and *SREBF1-1* (*Figure 4D*). In all, these results indicate that the expression of *PPARG* and *SREBF1* could be recapitulated in SC preadipocytes from obese IR/T2D patients by *PRPF8* complementation. In line with these findings, confocal microscopy studies showed that

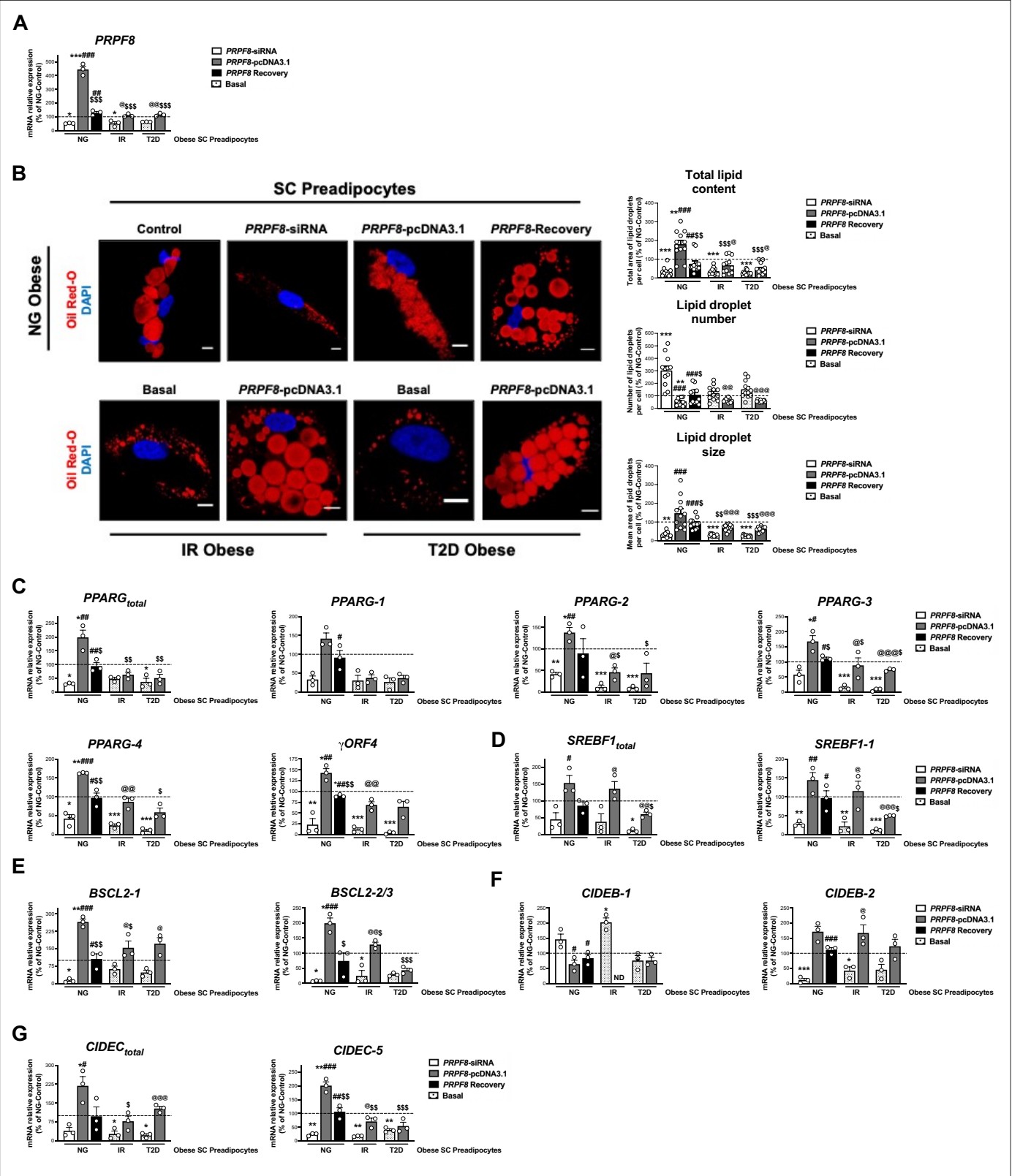

**Figure 4.** *PRPF8* silencing alters differentiation of human primary SC preadipocytes. Data corresponds to individuals from cohort 1. (**A**) Quantification of *PRPF8* mRNA levels in subcutaneous (SC) preadipocytes from normoglycemic (NG), insulin resistant (IR), and with type 2 diabetes (T2D) morbidly obese individuals transfected with control, *PRPF8*-siRNA, or *PRPF8*-pcDNA3.1, alone or in combination (*PRPF8* Recovery). (three replicate studies, 4 technical replicates each). (**B**) Representative confocal micrographs of SC preadipocytes from NG, IR, and T2D morbidly obese individuals after transfection

*Figure 4 continued on next page*

*Figure 4 continued*

with control, *PRPF8*-siRNA, or *PRPF8*-pcDNA3.1 alone or in combination (*PRPF8* Recovery) stained with Oil Red-O [lipid droplets (LDs), red] and DAPI (nucleus, blue). Morphometric analysis of LDs was carried out using ImageJ software. Scale bar = 10 μm. (12 replicate studies, 10 cells each). mRNA levels of *PPARG* (**C**), and *SREBF1* (**D**), and the LD-related proteins, *BSCL2* (**E**), *CIDEB* (**F**), and *CIDEC* (**G**) and their splicing forms in SC preadipocytes from NG, IR, and T2D morbidly obese individuals after transfection with control, *PRPF8*-siRNA or *PRPF8*-pcDNA3.1, alone or in combination (*PRPF8* Recovery). (three replicate studies, 4 technical replicates each). *p < 0.05, **p < 0.01, ***p < 0.001 *vs.* NG-Control; #p < 0.05, ##p < 0.01, ###p < 0.001 *vs.* *PRPF8*-siRNA; $p < 0.05. $$p < 0.01, $$$p < 0.001 *vs.* *PRPF8*-pcDNA3.1; @p < 0.05, @@p < 0.01, @@@p < 0.001 *vs.* corresponding basal. Data are presented as mean ± standard error of the mean (S.E.M.). One-way ANOVA with Tukey's multiple comparisons test or Kruskal-Wallis with Dunn's multiple comparisons test (for parametric or non-parametric data, respectively) were used. Normality distribution was determined by Shapiro-Wilk normality test.

*PRPF8*-pcDNA3.1 expression increased total lipid content in SC preadipocytes from obese subjects with IR/T2D by increasing LD size, while decreasing LD number (*Figure 4B*). Analysis of markers of LD biogenesis and growth (*BSCL-2*, *CIDEB*, *CIDEC*), showed similar trends to those observed for *PPARG* and *SREBF1* upon manipulation of *PRPF8* expression in SC preadipocytes from either NG or IR/T2D obese individuals (*Figure 4E–G*).

## The UPR is altered in preadipocytes of IR/T2D obese subjects

As mentioned earlier, pathway analysis of iTRAQ proteomic data indicated that ER stress-related pathways were altered in both SC and OM preadipocytes from T2D obese individuals when compared to NG obese subjects (*Figure 1B and C*).

Immunoblotting studies of additional human preadipocyte samples to those employed for iTRAQ studies, confirmed and extended the proteomic data by demonstrating altered mRNA/protein levels of UPR components in both SC and OM preadipocytes from IR/T2D obese groups as compared to NG obesity (*Figure 1B and C*, and *Figure 5*). To be more specific, SC and OM preadipocytes from T2D obese individuals exhibited enhanced levels of the ER stress-inducible gene, BiP (*Figure 5A* and *Figure 1—source data 2*). IR/T2D modified the protein content of p90 and p50, and cleaved/full-length ATF6 ratio in OM preadipocytes, and decreased this ratio in SC preadipocytes in IR *vs.* NG (*Figure 5B*). In addition, pPERK and pPERK/PERK ratio were up-regulated in SC and OM preadipocytes in relation to IR and/or T2D (*Figure 5C*). However, peIF2α and peIF2α/eIF2α ratio as well as the eIF2α-target, CHOP, were significantly reduced in OM preadipocytes in relation to IR/T2D (*Figure 5C and E*). We observed enhanced IRE1α levels and decreased pIRE1α/IRE1α ratio in IR *vs.* NG OM preadipocytes (*Figure 5F*). IR OM preadipocytes and IR/T2D SC preadipocytes also exhibited higher levels of the spliced form of *XBP1* than NG preadipocytes (*Figure 5G*). Finally, the ER chaperones, GRP94 and PDI, were more abundant in T2D preadipocytes than in NG preadipocytes from both SC and OM fat (*Figure 5H*). Our proteomic study revealed enhanced levels of 11 additional ER chaperones in T2D SC preadipocytes (*Figure 1—source data 2*).

## Dysregulation of ER-associated protein degradation (ERAD) in IR/T2D obese preadipocytes

In association with the UPR, the ERAD represents a key quality-control machinery that recruits unfolded/misfolded ER proteins via ER chaperones and targets these proteins for cytosolic degradation by the proteasome (*Christianson and Ye, 2014*; *Qi et al., 2017*). Given our results on the UPR, we next explored the ERAD in human preadipocytes and observed a marked up-regulation of proteins involved in all the steps comprising this process, including protein recognition (*BIP*; *Figure 6A*), retro-translocation through the ER membrane (*DERL1*, *SEC61A1*, *STT3A*, and *STT3B*; *Figure 6B*), ubiquitination (*HRD1* and *RNF185*; *Figure 6C*), and targeting of misfolded proteins to the proteasome (*RAD23A* and *UBQLN1*; *Figure 6D*) in IR/T2D *vs.* NG obesity. These changes occurred in both SC and OM preadipocytes, which split into two clusters (NG and IR/T2D) when ERAD data was represented in a two-way hierarchical clustering heatmap (*Figure 6E*). In this line, when the ERAD process was explored during adipocyte differentiation, higher overall transcript contents of ERAD genes in IR preadipocytes than in NG preadipocytes (measured as AUC) were observed (*Figure 6—figure supplement 1*).

Finally, when preadipocytes from individuals of cohort 2 (*Supplementary file 2*) were examined, a trend to increase in representative genes of ERAD in T2D individuals with simple obesity *vs.* their NG counterparts and lean individuals was observed (*Figure 6—figure supplement 2*). Nevertheless, the

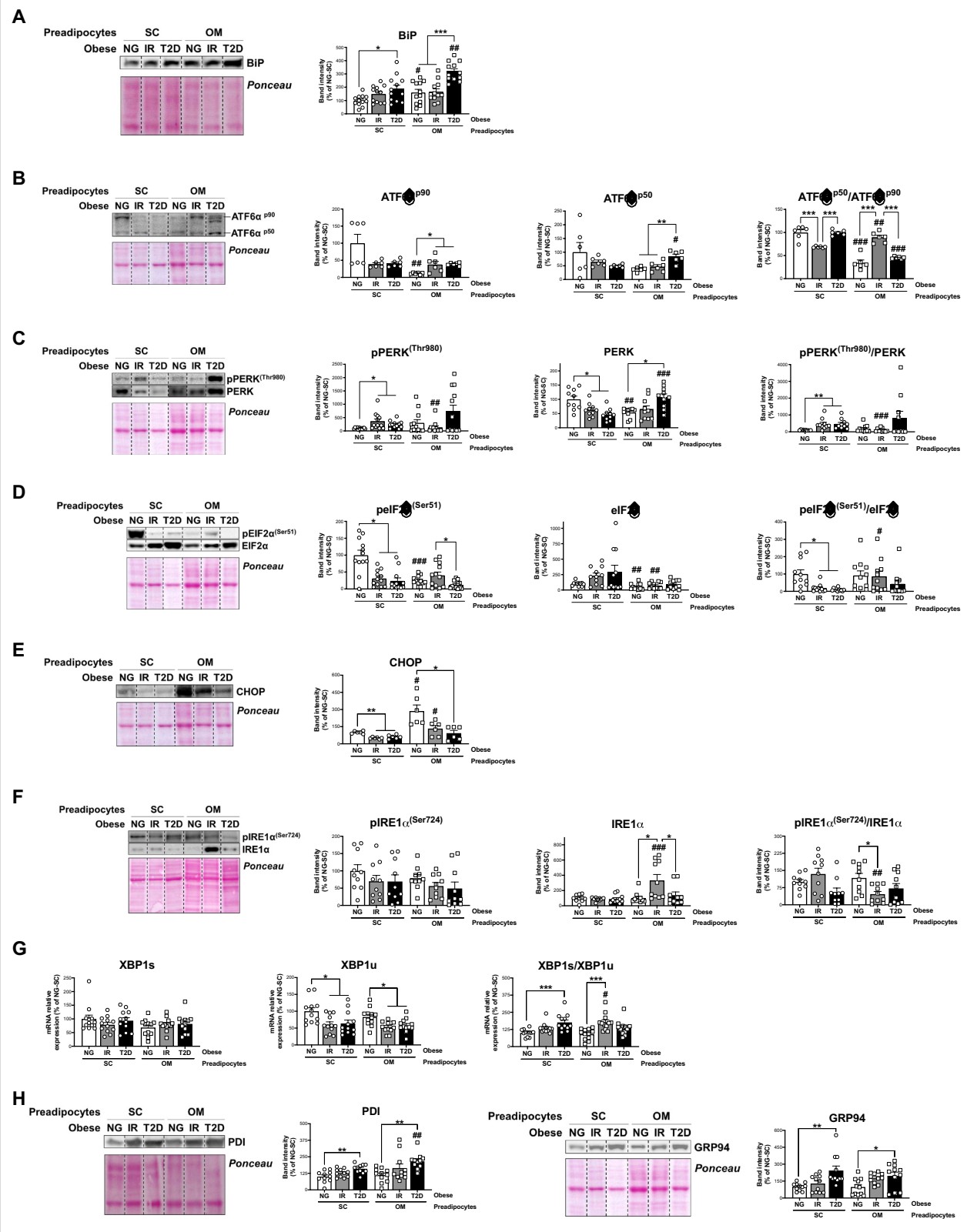

**Figure 5.** Unbalanced Unfolded Protein Response (UPR) in preadipocytes is associated with obesity-related insulin resistance (IR) and type 2 diabetes (T2D). Data corresponds to individuals from cohort 1. Representative blots and protein level quantifications of BiP (**A**), ATF6α$^{p90}$ and ATF6α$^{p50}$ (**B**), pPERK$^{(Thr980)}$ and PERK (**C**), pEIF2α$^{(Ser51)}$ and EIF2α (**D**), CHOP €, pIRE1α$^{(Ser724)}$ and IRE1α (**F**), and PDI and GRP94 (**H**) in subcutaneous (SC) and omental (OM) preadipocytes from normoglycemic (NG), IR and T2D morbidly obese subjects ( = 6–12, 1 technical replicate each). (**G**) mRNA levels of *XBP1s* and

*Figure 5 continued on next page*

*Figure 5 continued*

*XBP1u* in SC and OM preadipocytes from NG, IR and T2D obese subjects ( = 12, 1 technical replicate each). *p < 0.05, **p < 0.01, ***p < 0.001 *vs.* NG and/or IR subjects; #p < 0.05, ##p < 0.01, ###p < 0.001 *vs.* SC preadipocytes from the same subjects. Data are presented as mean ± standard error of the mean (S.E.M.). One-way ANOVA with Tukey's multiple comparisons test or Kruskal-Wallis with Dunn's multiple comparisons test (for parametric or non-parametric data, respectively, determined by Shapiro-Wilk normality test) were used.

expression of some of these ERAD-related genes, such as *RAD23A*, which was higher in SC preadipocytes from NG individuals with simple obesity as compared to lean individuals, decreased in the transition from NG to T2D in obesity (*Figure 6—figure supplement 2D*), an effect that was also observed in morbid obesity (*Figure 6D*).

## Regulation of ER proteostasis in preadipocytes

In order to unveil the regulation of ERAD in preadipocytes, in vitro models of hyperglycaemia/hyperinsulinemia, inflammation, and hypertrophy due to lipid overload, were developed using SGBS cells (*Figure 7—figure supplement 1*). Specifically, preadipocytes were exposed to high concentrations of glucose and insulin (HGHI), TNFα, or fatty acids (palmitate or oleate), respectively (*Díaz-Ruiz et al., 2015*). None of the treatments compromised cell viability (data not shown) and, except for oleate, they impaired insulin-induced Akt phosphorylation (*Figure 7—figure supplement 1C*). Exposure to HGHI increased the expression of BiP and nearly all the other ERAD components tested (*Figure 7—figure supplement 2A*). As shown in *Figure 7—figure supplement 2B*, expression levels of ERAD genes enabled discrimination of HGHI-treated from control SGBS cells. Given these observations, we next examined the activity of the protein degradation machinery in the cytosol, the proteasome (*Bard et al., 2018*), in cells exposed to HGHI. These studies showed that hyperglycaemic/hyperinsulinemic conditions decreased the activity of the 26 S proteasome while increasing the amount of ubiquitinated proteins in SGBS preadipocytes (*Figure 7—figure supplement 2C and H*).

The naturally occurring bile acid, tauroursodeoxycholic acid (TUDCA), has been shown to restore ER homeostasis in ER-stressed cells (*Zhang et al., 2018*). We thus tested whether exposure of SGBS preadipocytes to TUDCA prior to HGHI treatment could prevent HGHI-induced up-regulation of ERAD genes and accumulation of ubiquitinated proteins in these cells, which was proven to be the case (*Figure 7—figure supplement 2D and E*). TUDCA also reverted HGHI inhibitory effects on PPARγ and FABP4 expression levels (*Figure 7—figure supplement 2F*).

Finally, TUDCA was also able to reduce the enhanced expression levels of ERAD-related genes and decreased mRNA levels of adipogenic markers induced by BiP overexpression in SGBS preadipocytes (*Figure 7—figure supplement 3*).

Based on the results obtained in SGBS preadipocytes, we next tested whether exposure to HGHI could induce an IR-like phenotype in primary preadipocytes from OM adipose tissue of NG obese individuals (i.e., activation of ERAD). *Figure 7* shows that, as for SGBS preadipocytes, exposure of OM preadipocytes from NG obese subjects to HGHI increased the expression levels of genes involved in all ERAD steps (recognition, retrotranslocation ubiquitination, and targeting to the proteasome) in these cells (*Figure 7A–D*). On the other hand, TUDCA, which did not alter essentially the expression of ERAD genes when administered alone, reduced HGHI-induced gene expression increases to control levels (i.e. OM preadipocytes from NG obese individuals exposed to medium alone) (*Figure 7A–D*). HGHI also increased the accumulation of ubiquitinated proteins in OM preadipocytes from NG obese individuals, while only a slight, not significant decrease in ubiquitin-conjugated proteins was observed when HGHI was combined with TUDCA (*Figure 7E*). Notably, the activity of the 26 S proteasome remained unchanged in all the experimental conditions tested (*Figure 7F*).

Morphometric quantification of micrographs from HGHI-treated OM preadipocytes revealed a decrease in total lipid content as compared to their control counterparts (*Figure 7G*). These changes were accompanied by significant reductions in the expression of total *PPARG* and *PPARG-2* as well as of *ADIPOQ* and *FABP4* mRNA levels (*Figure 7H and I*). Exposure to TUDCA reverted the effects of HGHI on both lipid content and on most of the adipogenic genes examined (*Figure 7G–I*).

On the other hand, analysis of OM preadipocytes from both IR and T2D obese individuals demonstrated that proteasome activity was decreased while levels of ubiquitin-conjugated proteins were increased in these cells as compared to their NG counterparts (*Figure 7E and F*). Interestingly, TUDCA reverted both effects in OM preadipocytes from IR obese patients but not in T2D preadipocytes

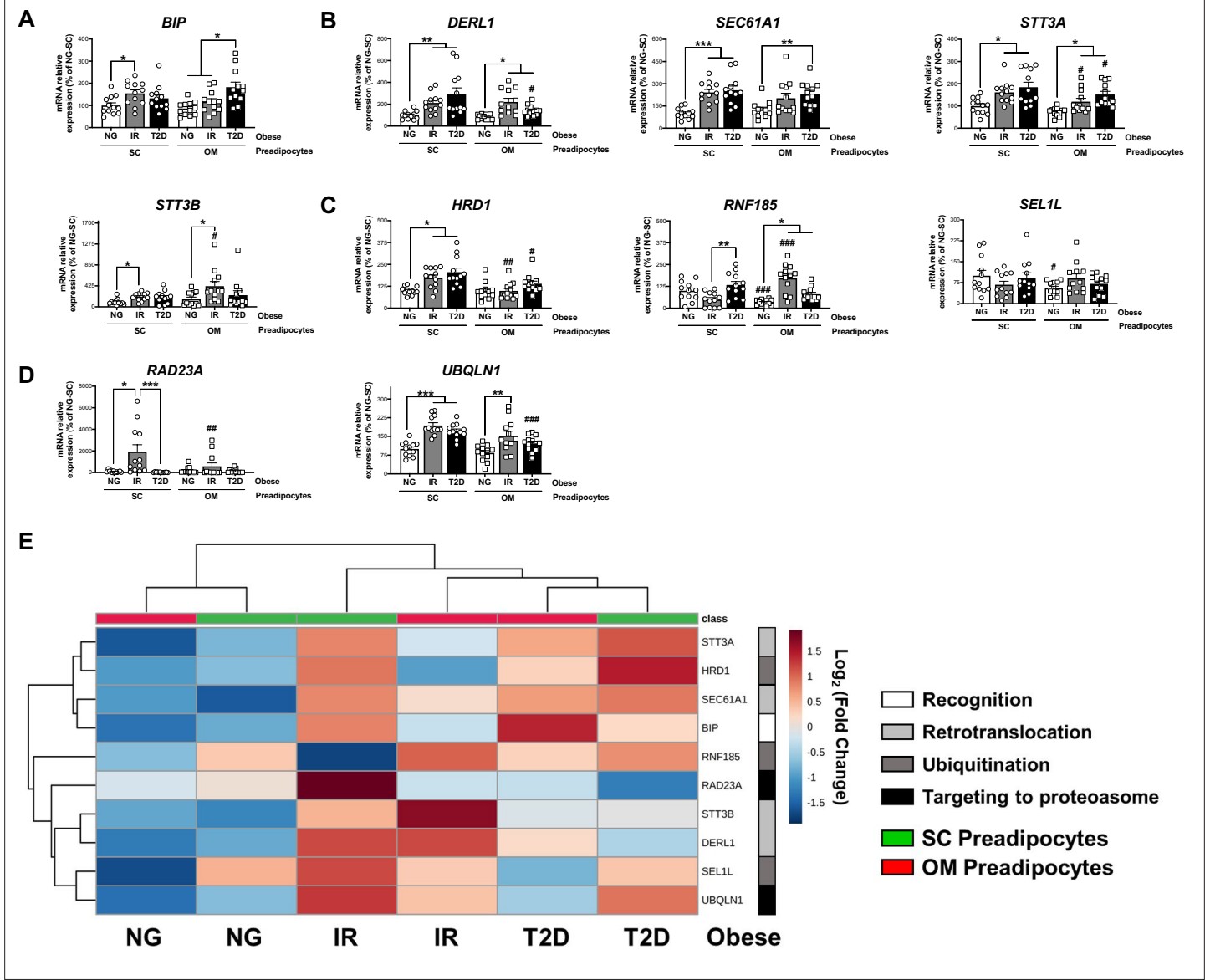

**Figure 6.** Activation of Endoplasmic Reticulum-Associated Degradation (ERAD) in preadipocytes is associated with obesity-related insulin resistance (IR) and type 2 diabetes (T2D). Data corresponds to individuals from cohort 1. mRNA levels of *BIP* (**A**), *DERL1*, *SEC61A1*, *STT3A* and *STT3B* (**B**), *HRD1*, *RNF185* and *SEL1L* (**C**), and *RAD23A* and *UBQLN1* (**D**), in subcutaneous (SC) and omental (OM) preadipocytes from normoglycemic (NG), IR and T2D morbidly obese subjects (n = 12, 1 technical replicate each). *p < 0.05, **p < 0.01, ***p < 0.001 *vs.* NG and/or IR subjects; #p < 0.05, #p < 0.01, ###p < 0.001 *vs.* SC preadipocytes from the same subjects. (**E**) Hierarchical clustering dendrogram heatmap analysis of ERAD-related genes in SC (green) and OM (red) preadipocytes from NG, IR, and T2D obese subjects. Rows stand for ERAD-related steps (white, recognition; light grey, retrotranslocation; dark grey, ubiquitination; black, targeting to proteasome), while columns stand for subject groups. The scale in the colour bar represents -Log$_2$(Fold Change). One-way ANOVA with Tukey's multiple comparisons test or Kruskal-Wallis with Dunn's multiple comparisons test (for parametric or non-parametric data, respectively, determined by Shapiro-Wilk normality test) were used. The online version of this article includes the following figure supplements for *Figure 6—figure supplement 1* and *Figure 6—figure supplement 2*.

The online version of this article includes the following figure supplement(s) for figure 6:

**Figure supplement 1.** mRNA levels of ERAD-related genes during in vitro differentiation of SC and OM preadipocytes from NG and IR morbidly obese individuals (cohort 1; n = 6, 1 technical replicate each).

**Figure supplement 2.** mRNA levels of representative components of the ERAD pathway, participating in the recognition (*BIP*) (**A**), retrotranslocation through the endoplasmic reticulum (ER) membrane (*DERL1* and *SEC61A1*) (**B**), ubiquitination (*RNF185*) (**C**), and targeting of misfolded proteins to the proteasome (*RAD23A*) (**D**) in SC and OM preadipocytes from lean individuals and NG and T2D subjects with simple obesity (cohort 2; n = 5–11, 1 technical replicate each).

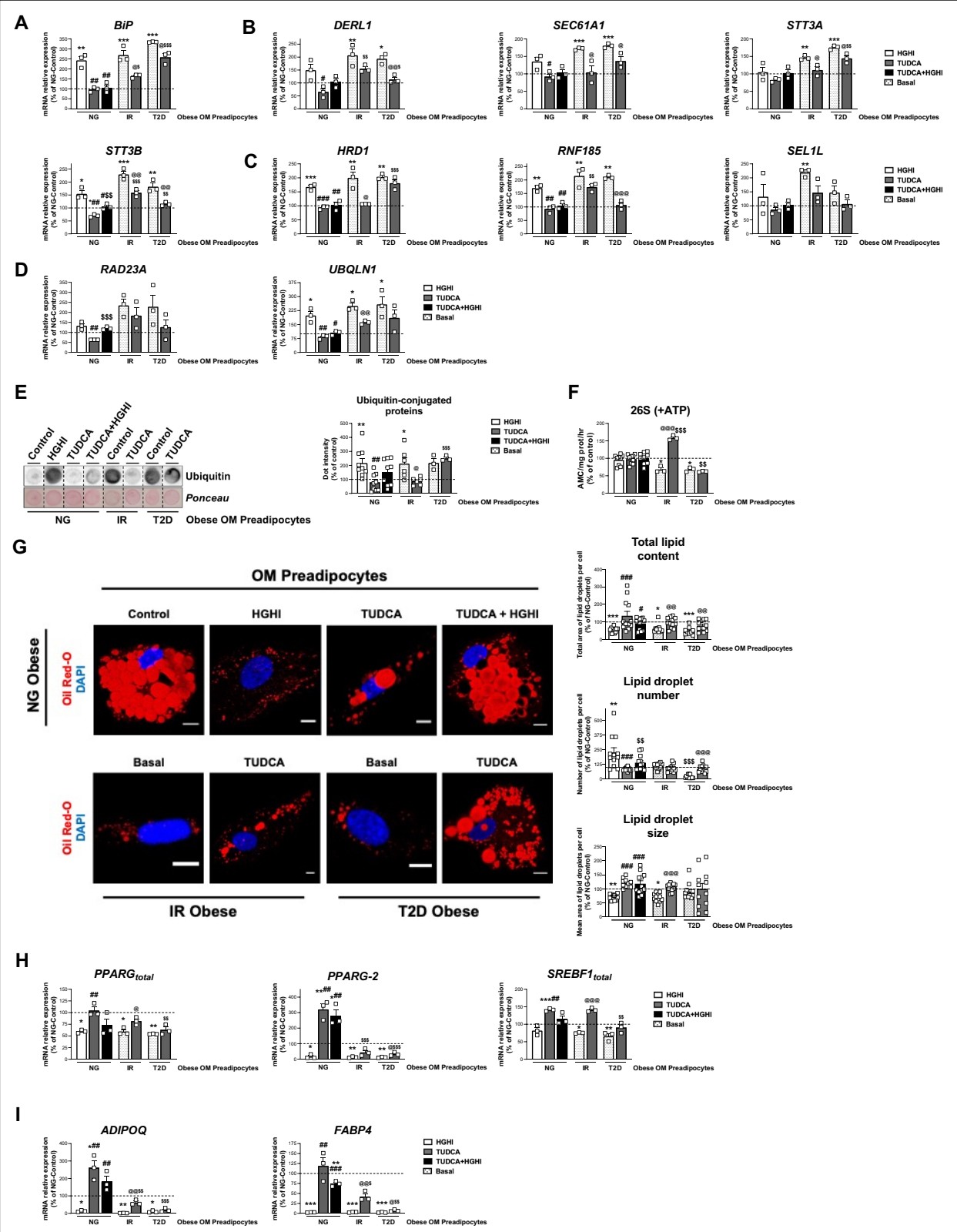

**Figure 7.** Activation of ERAD in human primary omental (OM) preadipocytes from normoglyceimc (NG) morbidly obese individuals by hyperglycemic / hyperinsulinemic (HGHI) conditions and reversal of ERAD activation in human primary OM preadipocytes from insulin resistant (IR) and with type 2 diabetes (T2D) morbidly obese individuals by TUDCA. Data correspond to individuals from cohort 1. mRNA levels of *BIP* (**A**), *DERL1*, *SEC61A1*, *STT3A*, and *STT3B* (**B**), *HRD1*, *RNF185* and *SEL1L* (**C**), and *RAD23A* and *UBQLN1* (**D**). (three replicate studies, 4 technical replicates each). Representative

*Figure 7 continued on next page*

*Figure 7 continued*

dot-blot and protein quantification of ubiquitin-conjugated proteins (**E**), and chymotrypsin-like peptidase activity of the 26 S proteasome (+ ATP) (**F**) in OM preadipocytes from NG, IR, and T2D morbidly obese individuals exposed 14 hr to 0.5 mg/mL TUDCA, 24 hr to HGHI conditions, and/or a combination of both. (3–9 replicate studies, 3 technical replicates each). (**G**) Representative confocal micrographs of OM preadipocytes from NG, IR, and T2D morbidly obese individuals under the indicated experimental conditions stained with Oil Red-O [lipid droplets (LDs), red] and DAPI (nucleus, blue). Morphometric analysis of LDs was carried out using ImageJ software. Scale bar = 10 μm. (12 replicate studies, 10 cells each). (**H–I**) mRNA levels of adipogenesis-related genes in OM preadipocytes from NG, IR and T2D morbidly obese individuals under the indicated experimental conditions. (three replicate studies, 4 technical replicates each). *p < 0.05, **p < 0.01. ***p < 0.001 *vs.* control; #p < 0.05, ##p < 0.01, ###p < 0.001 *vs.* HGHI; $ < 0.05, $$p < 0.01 *vs.* TUDCA; @p < 0.05, @@p < 0.01, @@@p < 0.001 *vs.* corresponding basal. Data are presented as mean ± standard error of the mean (S.E.M.). One-way ANOVA with Tukey's multiple comparisons test or Kruskal-Wallis with Dunn's multiple comparisons test (for parametric or non-parametric data, respectively) were used. Normality distribution was determined by Shapiro-Wilk normality test. The online version of this article includes the following figure supplements for *Figure 7—figure supplement 1*, *Figure 7—figure supplement 2*, and *Figure 7—figure supplement 3*.

The online version of this article includes the following figure supplement(s) for figure 7:

**Figure supplement 1.** Representative phase-contrast microscopy (x100 magnification and zoom) and confocal micrographs of SGBS preadipocytes exposed during 30 hr to a combination of glucose (4.5 g/L) and insulin (100 nM) (HGHI; white bars), 5 nM TNFα (grey bars), 500 μM Oleate (dark grey bars), or 500 μM Palmitate (black bars) (**A**).

**Figure supplement 2.** Regulation of Endoplasmic Reticulum-Associated Degradation (ERAD).

**Figure supplement 3.** mRNA levels of ERAD (**A**) and adipogenesis-related genes (**B**) in SGBS preadipocytes overexpressing BiP (pCMV-BIP, black), exposed 14 h to 0.5 mg/mL TUDCA (TUDCA, grey), or a combination of both (TUDCA + pCMV BIP, white with black dots) (three replicate studies, 4 technical replicates each).

(*Figure 7E and F*). TUDCA was also able to increase the expression of markers of differentiation in preadipocytes from IR and, to a lower extent, also from T2D obese individuals (*Figure 7H and I*). In fact, this chaperone enhanced the lipid content in OM preadipocytes of IR and T2D obese subjects to the levels found in OM preadipocytes of NG obese subjects, though TUDCA-treated T2D OM preadipocytes exhibited low LD numbers (*Figure 7G*).

## Discussion

We found that alternative splicing in the SC adipose tissue, and the ER stress-UPR-ERAD system in both SC and OM fat represent essential components of the adipogenic process that are altered in obese individuals with IR/T2D. We show novel evidence demonstrating that NG obesity involves the up-regulation of splicing-related genes in SC preadipocytes as compared with lean individuals. However, the development of IR/T2D, both in simple and morbid obesity, is associated with a general decrease in the expression of splicing genes in SC preadipocytes. In fact, decreasing the expression of a key component of the major spliceosome, *PRPF8*/PRP8, by siRNA is sufficient to hamper the expression of both markers of adipogenesis and regulators of LD biogenesis and growth. Our results support a relevant role for this protein and the splicing machinery in the regulation of adipocyte differentiation and highlight the biological significance of alternative splicing in maintaining adipose tissue homeostasis and metabolic health in obesity. Our results are also indicative of IR/T2D-dependent ERAD hyperactivation in preadipocytes of the two major fat depots, SC and OM, in obese individuals. Maladaptive UPR and ERAD signalling upon ER stress induction may contribute to impair adipocyte differentiation and to the progression of metabolic disease in obesity.

Alternative splicing of mRNA enables cells to acquire protein diversity, which is essential for cell differentiation and tissue growth and identity (*Baralle and Giudice, 2017*). It is accepted that both mRNA transcription and processing contribute to the changes in gene expression patterns leading to the transition of progenitor cells toward a mature phenotype (*Fiszbein and Kornblihtt, 2017*). In this regard, recent reports suggest a role for alternative splicing in adipocyte differentiation (*Lin, 2015*). It has been shown that inhibition of the splicing factor, SF3B1, blocks 3T3-L1 adipocyte differentiation (*Kaida, 2019*), and recent RNA-seq studies support the occurrence of alternative splicing events during adipogenesis of bonemarrow-derived human mesenchymal stem cells (*Yi et al., 2020*). Furthermore, certain naturally PPARγ isoforms generated by alternative splicing act as dominant-negative isoforms that prevent adipocyte differentiation (*Aprile et al., 2020*; *Aprile et al., 2018*; *Aprile et al., 2014*). However, few studies have addressed the relevance of the splicing machinery in human preadipocytes and whether this system is altered in obesity. To the best of our knowledge,

this is the first study demonstrating the deregulation of representative members of the two main components of the splicing machinery, the major and minor spliceosome, and associated splicing factors in human SC preadipocytes (and, occasionally, also in OM preadipocytes) in relation to obesity-associated IR/T2D. To be more specific, we show the up-regulation of several splicing-related genes in response to simple obesity, especially in SC preadipocytes, in individuals that maintain an adequate glycaemic control. This could represent an adaptive response to weight gain that seems to be lost in obese or extremely obese individuals with IR/T2D. In fact, our proteomic and gene expression results indicate that a high proportion of splicing-related elements were down-regulated in SC preadipocytes from obese individuals with IR/T2D as compared to those with a better metabolic profile (NG). In all, our results strongly support the notion that the splicing process is altered in obesity and related metabolic disease.

To analyse the relevance of alternative splicing on adipogenesis, we reduced the expression of *PRPF8*/PRP8 in SC preadipocytes from NG obese individuals to the levels found in their counterparts in IR/T2D obese individuals as well as in the human preadipocyte cell line, SGBS cells. In this context, recent work combining RNA-seq and proteomics demonstrated that PRP8 depletion altered the expression of more than 1,500 proteins in Cal51 breast cancer cells, supporting the relevance of this protein in defining the human proteome (*Liu et al., 2017*). In fact, as part of the core of the major spliceosome, PRP8 is potentially involved in the removal of most introns from precursor mRNAs (99.5 % of non-coding regions are removed by the major spliceosome) (*Turunen et al., 2013*). Our studies showed that decreasing PRP8 protein levels in NG obese SC preadipocytes or in SGBS preadipocytes evoked gene/protein expression changes indicative of altered adipogenesis, which was confirmed by the decreased accumulation of lipids observed in both cell types at late stages of differentiation. These results were replicated in another SC adipocyte cell line, hADSCs, derived from a lean donor, reinforcing the key contribution of *PRPF8*/PRP8 to human adipocyte differentiation.

Two different yet functionally interrelated pathways were impaired in either NG obese SC preadipocytes, SGBS preadipocytes or hADSCs cells silenced for *PRPF8* in terms of both total gene expression levels and isoform balance of the proteins involved, that is, adipogenic transcriptional program, and LD biogenesis and growth (*Figure 8*). Thus, PRP8 down-regulation altered the expression of PPARγ and SREBP1, as well as of seipin and two members of the CIDE family (CIDEB, CIDEC). In particular, the overall decrease in PPARγ and SREBP1 isoforms observed upon PRP8 down-regulation may account for the reduction observed in mRNA and/or protein levels of known downstream target genes of these transcription factors (adiponectin, FABP4, CIDEC, and FAS) (*Lowe et al., 2011*; *Slayton et al., 2019*). Likewise, the dramatic changes in LD size and number observed in *PRPF8*-silenced adipocytes might be related to the low expression levels of the canonical seipin variant, seipin1 (*Craveiro Sarmento et al., 2018*), observed in these cells. Indeed, our observations are in line with previous data on seipin knockout A431 cells, which also exhibited more numerous but smaller LDs than controls (*Salo et al., 2016*). Interestingly, seipin knock-down has been shown to suppress *PPARG* and *SREBF1* expression (*Fei et al., 2011*), thus suggesting that the phenotype of cells silenced for PRP8 may be the result of the action of this spliceosome component on multiple key genes contributing to adipogenesis and lipogenesis. In line with this notion, analysis of CLIP-sep data in the ENCORI database revealed the presence of PRP8 binding sites in *PPARG*, *SREBP1*, *BSCL2*, *CIDEB*, and *CIDEC* mRNAs. Further research is needed to fully establish the splicing targets for PRP8 in preadipocytes and to characterize the relevance of the resulting protein isoforms (and/or their ratio) in adipogenesis. Notwithstanding this, the observation that PRP8 re-expression in *PRPF8*-silenced SGBS cells and, most notably, in SC preadipocytes from IR/T2D obese individuals, can restore the expression and/or isoform balance of genes such as *PPARG*, *BSCL2*, *CIDEB*, or *CIDEC*, as well as normal LD content, supports a prominent role for this protein in lipid accumulation in adipocytes. Our data on *SFPQ* silencing and SF3B pharmacological inactivation would further extend this role to the whole splicing machinery. In this scenario, it is reasonable to propose that the decreased expression of splicing-related genes in SC preadipocytes in IR/T2D obesity as compared to NG obesity could be responsible, at least in part, of the loss of adipogenic capacity displayed by SC preadipocytes from obese individuals with metabolic disease. The regulatory mechanisms controlling the expression of splicing genes in preadipocytes under physiological and pathological conditions remain to be elucidated. Studies from our laboratory using in vitro models mimicking different obesogenic inputs (hyperglycaemia/hyperinsulinemia, adipocyte hypertrophy, inflammation) failed to modify *PRPF8* mRNA levels in preadipocytes (data

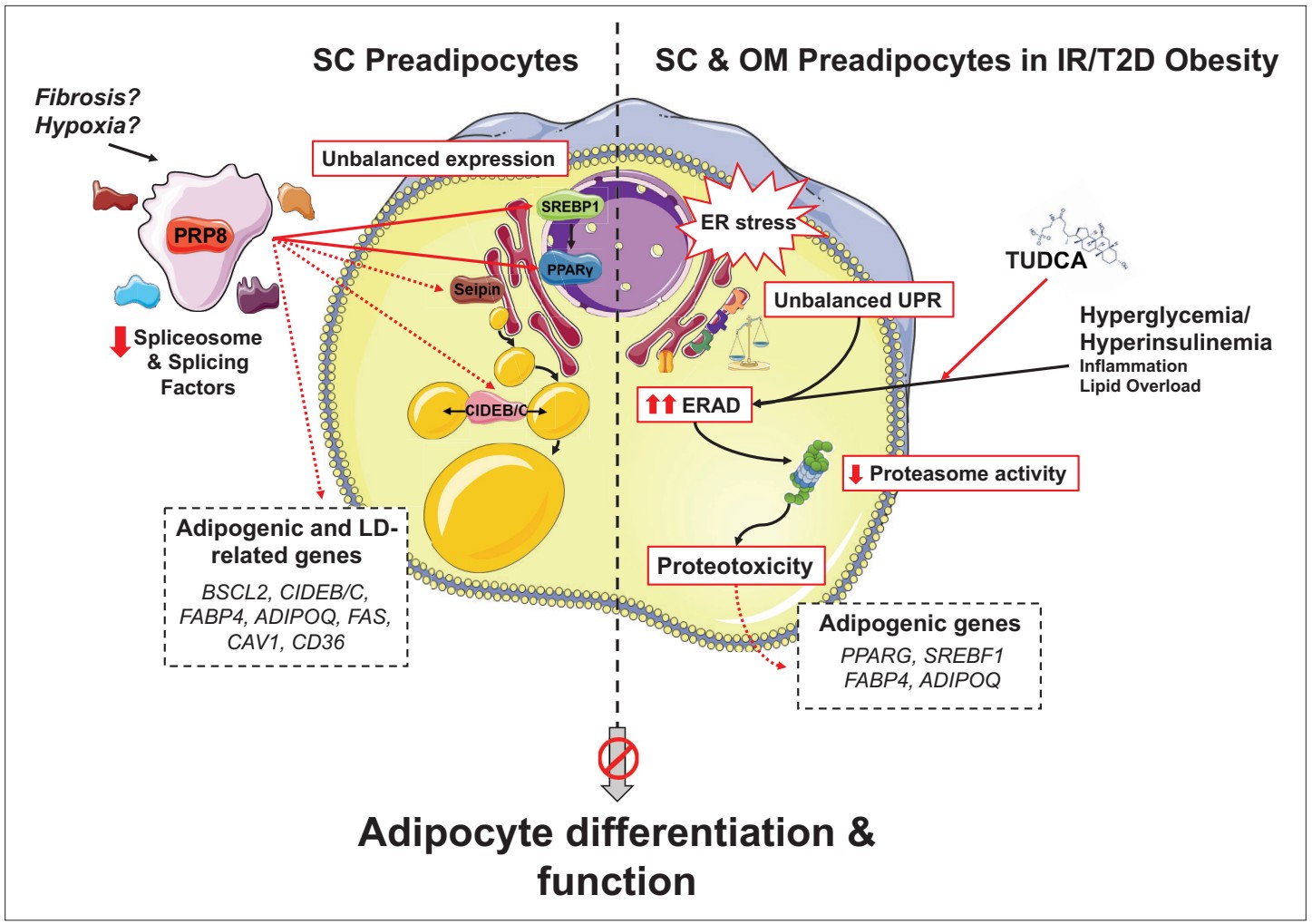

**Figure 8.** Schematic representation of the proposed mechanisms of action of PRP8/splicing and UPR/ERAD on adipocyte differentiation. PRP8 mediates alternative splicing events in subcutaneous (SC) preadipocytes by regulating the expression and/or isoform balance of the master regulators of adipogenesis, SREBP1 and PPARγ (solid arrows), likely through a direct action on the corresponding pre-mRNAs. This, in turn, would result in changes in the expression of adipogenic genes down-stream SREBP1 and PPARγ (dashed arrows), yet a direct action of PRP8 on CIDEB splicing is also plausible. The observed down-regulation of PRP8 in SC preadipocytes from obese individuals with insulin resistance (IR) and type 2 diabetes (T2D) would contribute to the impaired adipogenic capacity of these cells. A second pathogenic mechanism is activated in SC preadipocytes from IR/T2D obese individuals that impairs adipogenesis, that is, dysregulated ER proteostasis leading to UPR/ERAD activation. This mechanism, which is also triggered in omental (OM) preadipocytes under conditions of IR/T2D obesity, alters the adipogenic program by modifying the expression of key adipogenic transcripction factors and down-stream adipocyte markers. Splicing dysregulation and UPR/ERAD activation could be differentially triggered by obesogenic insults. The model is based on published molecular mechanisms, in silico analysis of ENCORI and HumanBase databases, and the main findings shown in this article. This figure was created using graphical elements from Servier Medical Art repository (SMART; https://smart.servier.com/).

not shown). Future studies will be aimed at establishing whether other pathogenic processes causing adipose tissue dysfunction in obese individuals (i.e. fibrosis, hypoxia) may alter alternative splicing in these cells (*Figure 8*).

When viewed together, our data strongly support the notion that splicing deregulation in SC preadipocytes may represent a pathogenic factor that contributes to the development of obesity-related complications. As mentioned earlier, changes in splicing components between groups were negligible in OM preadipocytes as compared to their SC counterparts. These inter-depot differences, which appear to be cell autonomous, might contribute, at least partly, to the distinct differentiation capacity of SC and OM preadipocytes observed in this and other studies (*Martyniak and Masternak, 2017*). Interestingly, along with altered splicing, SC preadipocytes in IR/T2D obesity also exhibited molecular features indicative of ER stress response failure, a hallmark process associated with the development of metabolic diseases (*Bhattarai et al., 2020*). Our data suggest that this pathogenic process also

occurs in OM preadipocytes in IR/T2D obesity. In accordance with these findings, previous studies from our laboratory have established that IR obesity is associated with enhanced levels of ER stress markers in mature adipocytes isolated from either SC or OM fat (*Díaz-Ruiz et al., 2015*). Herein we report, for the first time, that the three UPR signal transducers, IRE1α, PERK, and ATF6 (*Hetz and Papa, 2018*), and two effector systems downstream the UPR branches, CHOP and *XBP1* splicing, are dysregulated in primary SC and OM preadipocytes in the transition from NG to IR/T2D in severe obesity, in terms of total protein content and/or phosphorylation rate (i.e. activation). It is noteworthy that changes in these proteins did not always follow the same trends between SC and OM preadipocytes in response to IR/T2D, indicating the existence of depot-specific (mal)adaptive responses to obesity-associated ER stress. Notwithstanding this notion, a feature common to SC and OM preadipocytes was the up-regulation of both the master initiator of the UPR signalling pathway, BiP, and *XBP1s/ XBP1u* ratio, in IR/T2D obesity. Given the role of BiP in mediating the targeting of misfolded/ unfolded ER proteins for cytosolic degradation via ERAD, and the stimulatory effect of *XBP1s* on the expression of ERAD components (*Karagöz et al., 2019*), these observations supported that IR/T2D obesity could not only alter the UPR but also the activity of this branch of the quality-control system in the ER (*Hwang and Qi, 2018*). Interestingly, ERAD is activated when the protein folding capacity of the ER is exceeded (*Hwang and Qi, 2018*), which seems to be the case for SC and OM preadipocytes of IR/T2D morbidly obese individuals according to their enhanced expression levels of ER chaperones as compared to NG obesity. In fact, our studies show, for the first time, that SC and OM preadipocytes from IR/T2D obese individuals, as compared to NG obese individuals, display enhanced levels of proteins participating in all the steps comprising the ERAD, thus suggesting that this pathway is hyperactivated in preadipocytes during the progression to IR/T2D in obese individuals. Notably, our studies also revealed increased mRNA levels of genes involved in protein retrotranslocation from the ER to the cytosol in preadipocytes from individuals with simple obesity, especially in those with T2D, as compared to lean individuals.

In all, our data on the UPR and ERAD strongly support the notion that metabolic disease in obesity is characterized by the perturbation of ER proteostasis network components in preadipocytes. They also support the link between unresolved ER stress and adipogenesis dysregulation (*Longo et al., 2016*). Our studies using in vitro models mimicking different obesogenic inputs indicate that inflammation, adipocyte hypertrophy and, specially, hyperglycaemia/hyperinsulinemia (HGHI), as occurs in IR obese individuals, may contribute to alter ER proteostasis in preadipocytes. To be more specific, exposure to hyperglycaemic/hyperinsulinemic conditions induced ERAD hyperactivation, as well as BiP expression, in both OM preadipocytes from NG obese individuals and in SGBS preadipocytes. Intriguingly, HGHI-induced ERAD expression was accompanied by a concomitant increase in ubiquitinated proteins in preadipocytes and, in the case of SGBS cells, by a decrease in the activity of the cell machinery involved in misfolded/unfolded protein degradation, the proteasome. In this line are also our results on the decreased activity of the proteasome in OM preadipocytes from obese individuals with IR/T2D. These results suggest that the preadipocytes are unable to counteract the proteostatic stress occurring under IR/T2D conditions, and proteotoxicity may ensue. In agreement with these findings, we have reported the occurrence of proteotoxicity, that is, decreased activity of the proteasome and accumulation of ubiquitinated proteins, in in vitro differentiated preadipocytes and in mature adipocytes isolated from SC and OM fat of diet-induced obese mice as compared to lean animals, as well as in mature adipocytes isolated from IR *vs.* NG obese individuals (*Díaz-Ruiz et al., 2015*). In all, our studies strongly support the notion that dysregulated ERAD in preadipocytes represents a relevant pathological mechanism linked to the development of metabolic disease in obese individuals (*Figure 8*). In this line, altered ERAD expression levels in NG OM preadipocytes and SGBS cells exposed to HGHI were accompanied by changes in adipogenic markers demonstrative of impaired adipocyte differentiation. These effects were reverted by pre-exposure of these cells to the ER stress-reducing agent, TUDCA (*Zhang et al., 2018*), which also reduced the enhanced expression levels of ERAD genes found in OM preadipocytes of IR obese individuals or, although to a lesser extent, in T2D OM preadipocytes. Together, these findings suggest that TUDCA may represent a potentially useful treatment for preadipocyte damage prevention in obesity, as it has been proposed for this bile acid in other ER stress-related pathologies (*Kusaczuk, 2019*).

In summary, our results unveil an important role for alternative splicing in SC preadipocyte differentiation thus paving the way for the development of novel therapeutic strategies to modulate adipose

tissue expansion. Our studies on altered ER dynamics, that affect not only the UPR but also the ERAD pathway, in SC and OM preadipocytes are in line with our previous results in mature adipocytes (*Díaz-Ruiz et al., 2015*) and, in all, indicate that the maintenance of ER proteostasis in adipose tissue cells is key to ensure metabolic health in obesity.

## Materials and methods

### Subjects (Cohort 1)

A total of 78 morbidly obese subjects (BMI >40 kg/m$^2$) undergoing bariatric surgery were recruited at the General and Digestive Surgery Unit and the Lipids and Atherosclerosis Unit of the Reina Sofía University Hospital (HURS; Córdoba, Spain). Written consent was obtained from all the participants prior to recruitment, and the experimental protocol was approved by the Ethics and Research Committee of HURS following the Helsinki Declaration (Ethics Committee HURS, ref 3170). Anthropometric and biochemical parameters were obtained as described (*Díaz-Ruiz et al., 2015*). Subjects are of Caucasian origin, aged 18–60 years. Type 2 diabetes mellitus (T2D) in treatment with insulin, serious systemic disease not related to obesity (infectious disease, cancer, kidney disease, or severe liver disease) or major cardiovascular disease in the 6 months prior to the inclusion of the study, pregnancy, or breastfeeding and acute or chronic inflammatory diseases were considered as exclusion criteria.

Subjects were matched by age and BMI and subclassified into three groups [normoglycemic (NG Obese, n = 30;  15 females and  15 males): Glucose <100 mg/dL and HbA1c < 5.7%; impaired fasting glucose (IFG Obese, n = 30;  15 females and  15 males): Glucose 100–126 mg/dL and HbA1c 5.7%–6.4%; and diagnosed with type 2 diabetes (T2D Obese, n = 18; 11 females and  7 males): Glucose >126 mg/dL and HbA1c > 6.4%], according to the criteria of the American Diabetes Association (*ADA, 2021*). The clinical characteristics of the subjects are shown in *Table 1*. IFG individuals exhibited significantly higher HOMA-IR values than NG individuals, as well as other clinical and plasma parameters that are within the cut-off points for identifying insulin resistance in hyperinsulinemia-euglycemic clamp studies (*Tam et al., 2012*). Thus, they will be referred to hereinafter as insulin-resistant (IR) subjects (*Díaz-Ruiz et al., 2015*).

### Subjects (Cohort 2)

Forty-three subjects were recruited by the Endocrinology and Surgery departments at the University Hospital Joan XXIII (Tarragona, Spain) as reported previously (*Ejarque et al., 2017*). All subjects were of Caucasian origin and reported that their body weight had been stable for at least 3 months before the study. They had no systemic disease other than obesity, and all had been free of infection in the previous month before the study. Primary liver disease, cardiovascular disease, arthritis, acute inflammatory disease, infectious disease, neoplastic and renal diseases were specifically excluded by biochemical work-up. Subjects were classified by BMI according to World Health Organization criteria as lean (n = 18, BMI < 25 kg/m$^2$) and obese (n = 25, BMI > 30 kg/m$^2$). Obese subjects were subclassified into NG (n = 15) and T2D (n = 10) groups, as previously described (*Serena et al., 2016*). OM and SC adipose tissue samples were obtained during scheduled non-acute surgical procedures including laparoscopic surgery for hiatus hernia repair or cholecystectomies in non-morbid obese population. Anthropometric and biochemical variables from the cohort are presented in *Supplementary file 2*.

The hospital ethics committee approved the study and informed consent for biobanking surgically removed tissue was obtained from all participants in accordance with the Declaration of Helsinki.

### Adipose tissue processing

Paired abdominal subcutaneous (SC) and omental (OM) adipose tissue biopsies were obtained during bariatric surgery and processed as previously described (*Díaz-Ruiz et al., 2015*; *Peinado et al., 2010*). Samples were washed with Dulbecco's phosphate buffered saline (D-PBS) to remove blood contaminants and mechanically dispersed in DMEM/F-12 (1:1). Next, samples were enzymatically dispersed by incubation in DMEM/F-12 (1:1) containing 400 units/mL of collagenase type V at 37 °C for 30 min in a shaking bath. Undigested tissue was removed by filtering through a sterile 100 µm pore Cell Strainer (BDFalcon, Cat#352360) and the remaining was centrifuged at 600 x g for 10 min to separate the floating mature adipocyte layer and the pelleted stromal-vascular fraction (SVF). SVF was resuspended in DMEM/F-12 (1:1), filtered through a 40 µm pore Cell Strainer (BDFalcon, Cat#352340) and

centrifuged at 400 x g for 5 min. Then, pelleted SVF was resuspended in 500 µL of RBC Lysis Buffer and incubated for 3 min at room temperature (RT). After centrifugation at 400 x g for 10 min, SVF was frozen in liquid nitrogen and stored at –80 °C for further analysis and/or seeded onto culture flasks.

Freshly isolated mature adipocytes were washed with DMEM/F-12 (1:1) and added to lipolysis buffer [Krebs Ringer Phosphate (KRP) buffer: 0.9 % NaCl, 15 mM $NaH_2PO_4$, 6 mM KCl, 1.5 mM MgSO4, 1.6 mM $CaCl_2$ supplemented with 2 % bovine serum albumin (BSA), 1 mg/mL glucose and 0.1 mg/mL ascorbic acid; pH 7.4]. Fat cell volume and weight was determined as described (*Tchoukalova et al., 2003*) and discussed (*Lundgren et al., 2007*). In brief, light microscopy images of 100 cells were captured with a coupled camera (Moticam 1080; Motic, Barcelona, Spain) and their diameters (d) were measured in micrometres using ImageJ 1.50b. Mature adipocytes are assumed to be spheres so that their volume (expressed in picolitres) was calculated as ($[π×d^3]/6$) where d is the cell diameter in micrometres.

## Human primary preadipocytes

Human primary preadipocytes were cultured as previously described (*Serena et al., 2016*; *Guzmán-Ruiz et al., 2014*). SVF cells obtained from fresh SC and OM adipose tissue samples were seeded in preadipocyte-proliferation medium DMEM/F-12 (1:1) supplemented with 8 mM biotin, 18 mM d-pantothenate acid, 100 mM ascorbate, and 1 % penicillin-streptomycin, and 10 % new-born calf serum (NCS) at 37 °C in a humidified atmosphere with 95 % air: 5 % CO2. Medium was replaced every 48 h until confluence. Once in confluence ( > 80%), the cells were detached with trypsin- EDTA solution and subcultured at 4,000 cells/$cm^2$ 2–3 times to purify and amplify the cell culture following established methods (*Bunnell et al., 2008*; *Palumbo et al., 2018*; *Zhu et al., 2013*). Thereafter, preadipocytes were collected and frozen in liquid nitrogen and stored at –80 °C for immunophenotyping analysis, proliferation studies, or directly induced for adipogenic differentiation, as indicated in detail in the following sections.

## Immunophenotyping analysis

Freshly isolated SVF cells and cell cultures derived from SVF at passages 2–3 were processed for detection of preadipocyte markers and immune cell markers to assess the purity of the cell preparations employed for further analysis. To be more specific, the protein content of DLK1/PREF1 (preadipocyte marker), CD45 (leukocyte marker), and CD14 (macrophage marker) was assessed in extracts from the SVF and preadipocytes that were obtained from the same SC and OM adipose tissue samples following the procedures indicated in the 'Quantitative immunoblotting' section below.

## In vitro differentiated human adipocytes

Preadipocytes were seeded at 4000 cells/$cm^2$ in preadipocyte proliferation medium until they reached 70–80% of confluence. Then (day 0 of differentiation), primary preadipocytes were differentiated keeping them in differentiation medium: preadipocyte-proliferation medium with 3 % NCS and 17.5 mM glucose, supplemented with 10 µg/mL insulin, 0.1 µM dexamethasone, 1 µM rosiglitazone, and 0.5 mM IBMX for the first 3 days. Then, medium was removed and replaced by differentiation medium supplemented with 10 µg/mL insulin and 0.1 µM dexamethasone for four more days (day 6), when the medium was refreshed and maintained until day 10. The adipogenic process was monitored by the appearance of doubly refractile lipid inclusions by light microscopy and images of the cultures were taken with a coupled camera (Moticam 1080; Motic, Barcelona, Spain). Experiments were carried out using cells from passages 3–4.

## Human adipose tissue cells lines

In this study we employed SGBS cells, a human-derived preadipocyte cell line isolated from the SVF of SC adipose tissue from a 3 months male infant with Simpson-Gobali-Behmel syndrome, was kindly donated by Prof. Dr. Martin Wabitsch (Ulm University, Germany). This cell model has been largely used in studies pertaining to preadipocytes and differentiated adipocytes (*Allott et al., 2012*; *Fischer-Posovszky et al., 2008*; *Kalkhof et al., 2020*; *Newell et al., 2006*; *Zandbergen et al., 2005*). SGBS cells growth and differentiation into adipocytes were performed as previously described (*Moure et al., 2016*). Cells were seeded at a density of 4000 cells/cm2 and proliferated to 80 % confluence

in basal medium: DMEM/F12 (1:1) supplemented with 32.7 µM biotin, 16.78 µM d-panthothenic acid, and 1 % penicillin-streptomycin.

Confluent SGBS cells (day 0 of differentiation) were differentiated keeping them in differentiation medium: serum-free basal medium, supplemented with 0.01 mg/mL human transferrin, 20 nM human insulin, 100 nM hidrocortisone, 0.2 nM 3,3',5-triiodo-l-thyronine (T3), 25 nM dexamethasone, 0.5 mM IBMX, and 2 µM rosiglitazone for the first 4 days. Next, medium was removed and replaced by differentiation medium supplemented with medium 0.01 mg/mL human transferrin, 20 nM human insulin, 100 nM hidrocortisone, and 0.2 nM T3 for 6 days (day 10). Then, medium was refreshed, and cells were maintained in culture until day 14. The differentiation process was monitored by microscopic observation of LDs and expression analysis of known markers of adipogenesis and LD biogenesis (data not shown).

We also employed an adipose-derived stem cell line (hADSCs) that was isolated from SC adipose tissue of a male lean donor (aged 16 years, BMI 24 kg/m$^2$). This cell model was developed and validated in our previous works (*Acosta et al., 2017*; *Ehrlund et al., 2017a*; *Ehrlund et al., 2017b*; *Petrus et al., 2020*). Cells were isolated, propagated, and differentiated into adipocytes as described (*Gao et al., 2014*; *Gao et al., 2017*). Cells were cultured at 20,000 cells/cm$^2$ in proliferation medium [DMEM low glucose (1 g/L glucose), pyruvate supplemented with 1 % hepes buffer 1 M, 0.5 % penicillin/streptomycin (10,000 U/mL), and 10 % FBS] with 2.5 ng/mL human FGF2. Cells were then proliferated until 80 % of confluence and the medium was replaced for proliferation medium without FGF2 for one more day.

Cells at day 0 of differentiation were differentiated keeping them in differentiation medium [serum-free proliferation medium:Ham's F-12 Nutrient Mix (1:1)] supplemented with 5 µg/mL human insulin, 10 µg/mL transferrin, 0.2 nM T3, 1 µM rosiglitazone, 100 µM IBMX, and 1 µM dexamethasone for the first 3 days. Next, at day 3 of differentiation, medium was removed and replaced by differentiation medium supplemented with 5 µg/mL human insulin, 10 µg/mL transferrin, 0.2 nM T3, and 1 µM rosiglitazone. The medium was refreshed each 2–3 days until day 6 and day 10 of differentiation. Differentiation process were monitored by microscopic observations of LDs.

All the cells used tested negative for mycoplasma contamination using a specific commercially available kit.

## iTRAQ labelling and high-resolution LC-MS/MS

A total of 1 × 10$^6$ human SC and OM preadipocytes from NG and T2D obese subjects were homogenized in lysis buffer containing 8 M urea, 4 % CHAPS, 30 mM Tris base, sonicated, and quantified by RcDc assay kit. Reduction and alkylation were done by addition of 2 mM 1,4-dithiothreitol (DTT) and 7 mM iodoacetamide (IAA), respectively, for 15 min at RT. Then, samples from two to three individuals per group and fat depot were pooled and used for two separate iTRAQ experiments (n = 5–6 subjects per group and fat depot). Protein samples were precipitated using 10 % trichloroacetic acid (TCA), diluted in tetraethylammonium bromide (TEAB) with 0.1 % sodium dodecyl sulphate (SDS) and quantified. Samples were diluted 1:2 with TEAB and digested with 2 % trypsin. Tryptic peptides (240 µg per group and fat depot) were employed for proteomic analysis using 4-plex isobaric tags for iTRAQ according to the manufacturer's instructions. Samples were tagged with the corresponding iTRAQ reagent, mixed, desalted and fractionated by strong cation exchange (SCX).

Six fractions were collected, and each fraction was desalted, evaporated to dryness, and diluted in 20 µL of injection phase. Then, 8 µL of extract were diluted with 5 % methanol (MeOH)/1 % formic acid and analysed by liquid chromatography-tandem mass spectrometry (LC-MS/MS). The MS system used was an Orbitrap XL (Thermo Scientific) equipped with a microESI ion source (Proxeon; Madrid, Spain) and coupled to an Agilent 1,200 series LC-system (Agilent Technologies; Madrid, Spain). The SCX-fractions were loaded onto a chromatographic system consisting of a C18 preconcentrating cartridge (Agilent Technologies) connected to a 15 cm long, 100 µm i.d. C18 column (Nikkyo Technos Co., Ltd.; Tokyo, Japan). The separation was performed at 0.4 µL/min in a 90 min acetonitrile gradient from 3% to 40% [solvent A: 0.1 % formic acid, solvent B: acetonitrile with 0.1 % formic acid]. The Orbitrap XL was operated in the positive ion mode with a spray voltage of 2 kV. The scan range for full scans was m/z 400–1800. The spectrometric analysis was performed in a data dependent mode, acquiring a full scan followed by 8 MS/MS scans of the four most intense signals detected in the MS scan. For each MS signal, two MS/MS spectra were acquired using

higher energy collisional dissociation (HCD) and ion-trap-based collision-induced dissociation (CID) as fragmentation devices. The HCD spectra were used to measure the intensity of iTRAQ fragments and the CID spectra were used for database search and peptide identification. An exclusion time of 30 sec and a spectral count of 2 were included to avoid repetitive MS/MS analysis of the dominant MS signals.

## Database search and protein identification

Peak lists were searched against the complete human proteome (UniProt release 2014_08, with 68049 proteins) using SEQUEST (Proteome Discoverer 1.3, ThermoFisher) as search engine. The search tolerances were set as follows: peptide mass tolerance 10 ppm, fragment tolerance 0.8 Da, with trypsin as enzyme (allowing up to two missed cleavages), methionine oxidation ( + 15.995 Da) and 4-plex iTRAQ (K, Y, N-terminal, + 144.102) as variable modifications, and cysteine carbamidomethylation ( + 57.021 Da) as fixed modification. Peptide identifications were filtered for 0.5 % FDR and only proteins identified with two or more peptides were considered. All the raw data of mass spectrometry measurements, together with protein identification have been deposited to the ProteomeXchange Consortium via the PRIDE (*Perez-Riverol et al., 2019*) partner repository with the dataset identifier PXD015621.

## Data normalization and statistical analysis

Protein intensities, for each of the two iTRAQ experiments, were normalized using protein median intensities in each comparison (iTRAQ reagents 115/116: 0.987, 115/117: 0.958, 116/114: 1.142, 117/114: 1.191 for replicate 1; and 115/116: 1.026, 115/117: 0.933. 116/114: 1.090, 117/114: 1.180 for replicate 2). Protein identified in the two replicates were matched. To discard duplicate proteins between replicates, the following parameters were considered: absence of missing values, # unique peptides, # peptide and coverage (the higher, the better). Proteins showing quantification values within a fixed average ratio ( ± 1.5) and a coefficient of variation (CV) under 20 % were considered for further statistical analysis. Specifically, the Perl module 'Statistic R' was used to calculate the average, the standard deviation (SD), the coefficient of variation (CV), and the fold change ratios for each protein between groups. Data normality was tested by Shapiro-Wilk test, and a Student's t test was performed to obtain the statistical significance (p-*value*).

## Bioinformatics analysis

Gene Ontology (GO) analysis of the proteomic results was conducted using online open-source software PANTHER (Protein ANalysis THrough Evolutionary Relationships) classification system 14.1 that annotated proteins to biological processes (*Mi et al., 2019*). Significant over- and under-represented GO terms and pathways were identified by Fisher's Exact with FDR multiple test correction <0.05 taking all the *Homo sapiens* genes in the data set as a reference list. To determine quantitative changes in this study, a ± 1.5 fold change, with p < 0.05 (determined by t-test) in two replicates, was set as the threshold for categorizing up-regulated and down-regulated proteins. Pathway and upstream regulator analysis were performed using IPA (Ingenuity Pathways Analysis) software 49309495. Canonical pathway analysis identified the pathways from the IPA library of canonical pathways that were most significant to the data set. The significance of the association between the data set and the canonical pathway was measured in two ways: (1) A ratio of the number of proteins from the data set that mapped to the pathway divided by the total number of proteins that mapped to the canonical pathway displayed. (2) Fisher's exact test was used to calculate a p-value determining the probability that the association between the proteins in the dataset and the canonical pathway was explained by chance alone, and p < 0.05 was considered statistically significant. The authors thankfully acknowledge the computer resources, technical expertise and assistance provided by the PAB (Andalusian Bioinformatics Platform) centre located at the University of Málaga (http://www.scbi.uma.es). mRNA binding sites and adipose tissue-specific functional interactions of *PRPF8*, *SF3B1*, and *SFPQ* were investigated using ENCORI (The Encyclopedia of RNA Interactomes; http://starbase.sysu.edu.cn/index.php) (*Li et al., 2014*) or SpliceAid-F (http://srv00.recas.ba.infn.it/SpliceAidF/) (*Giulietti et al., 2013*) and HumanBase tool (https://hb.flatironinstitute.org/) (*Greene et al., 2015*), respectively.

## Silencing studies in preadipocytes

Primary human SC preadipocytes from NG obese individuals (cohort 1) and SGBS preadipocytes, cultured on 6-wells or 12-wells plates, were stably transfected at day 4 of differentiation with targeted double stranded siRNA oligonucleotides against the major spliceosome component, *PRPF8*, the splicing factor, *SFPQ*, or their corresponding specific negative control siRNA using Lipofectamine RNAiMAX Transfection Reagent following manufacturer's instructions. Transfection efficiency was checked by RT-PCR, western blotting and, when possible, by cotransfecting cells with N-terminal end of phrGFP-N1, as a reporter (*Moreno-Castellanos et al., 2017*) using Lipofectamine 2000 Transfection Reagent. Briefly, the cells were washed twice with D-PBS and submerged into 1 mL of OPTI-MEM medium containing 100 nM of the corresponding siRNA and, when required, 3 µg of phrGFP-N1 during 24 hr. Thereafter, the transfection medium was removed, and fresh culture medium was added. Finally, after 3–5 days of transfection, silenced cells were stained with Oil Red-O. The effects of *PRPF8* and *SFPQ* silencing were also explored by RT-PCR and quantitative immunoblotting. In another series of experiments, *PRPF8* siRNA-treated SGBS cells were exposed to 100 mM insulin for 15 min.

hADSCs were transfected one day before differentiation induction through electroporation by Kit Neon Transfection System (*Kim et al., 2008*). The cells were trypsinised and washed with D-PBS (without $Ca^{2+}$ and $Mg^{2+}$) and resuspended in Resuspension Buffer R at a final density of $1 \times 10^6$ cells/mL. The NEON system (Invitrogen) 10 µL electroporation tips were used with $10^6$ cells per reaction. One pulse (1200 V, 40 ms) was used to transfect a final concentration of 40 nmol/L siRNA to cells, which were seeded in medium without antibiotic/antimycotic. Twenty-four hr after electroporation, the medium was replaced, and differentiation started following the standard protocol. Finally, the effects of PRPF8 silencing were evaluated at different days of differentiation (0, 3, 6, and 10 days of differentiation).

## Overexpression studies in preadipocytes

Primary human SC preadipocytes from NG obese individuals (cohort 1) and SGBS preadipocytes, cultured on 6-wells or 12-wells plates, were stably transfected at day 3–4 of differentiation with a plasmid vector coding for *PRPF8* (*PRPF8*-pcDNA3.1), *SFPQ* (*SFPQ*-pcDNA3.1), their corresponding empty plasmid vector alone (pcDNA3.1) as a negative control, or co-transfected in combination with their corresponding siRNA using Lipofectamine 2000 Transfection Reagent following manufacturer's instructions. In another sets of experiments, SGBS preadipocytes were stably transfected with a plasmid vector coding for *BIP* (pCMV *BiP*-Myc-KDEL-wt) or its corresponding empty vector (pCMV-Myc), Transfection efficiency was checked by RT-PCR, western blotting and, when needed, by co-transfecting cells with N-terminal end of phrGFP-N1, as a reporter (*Allott et al., 2012*; *Moreno-Castellanos et al., 2017*). Briefly, the cells were washed twice with D-PBS and submerged into 1 mL of OPTI-MEM medium containing 3 µg of corresponding plasmid vector and 100 nM of the siRNA during 24 hr. Thereafter, the transfection medium was removed, and fresh culture medium was added. Finally, after 3–5 days of transfection, silenced cells were stained with Oil Red-O.

The effects of *PRPF8* overexpression were also explored in primary human SC preadipocytes from IR/T2D obese individuals (cohort 1) by RT-PCR and quantitative immunoblotting. These techniques were also employed to study the effects of BiP overexpression in SGBS preadipocytes. All constructs were verified by DNA sequencing.

## Lipid droplet (LD) morphometric analysis

Oil Red-O staining of cellular lipids was performed in human preadipocytes and SGBS cells grown on glass coverslips under different conditions as previously described (*Pulido et al., 2011*). Cells were washed with D-PBS and fixed with 4 % paraformaldehyde for 8 min at RT. Coverslips were washed with 60 % isopropanol and then left to dry completely. Subsequently, an Oil Red-O stock solution was prepared: 0.35 g Oil Red-O was dissolved in 100 mL 100 % isopropanol. Before use, the solution was diluted 6:4 with distilled water, allowed to stand for 10 min, and then filtered through Whatman no. 1 paper. Cells were exposed to this solution for 30 min at RT in darkness and the unbound dye was rinsed with distilled water. Images were captured with an inverted light microscope coupled to a camera. For morphometric studies of the lipid droplets (LDs), coverslips were mounted on slides with fluorescent mounting medium containing 1 µg/mL DAPI, to visualize the nuclei, and examined by confocal microscopy as described previously (*Pulido et al., 2011*). At least 30 cells were randomly

selected for each experimental condition and the LD number and size were then estimated by using ImageJ 1.50b (**Deutsch et al., 2014**; **Mohan et al., 2019**). Analysis was performed by converting the 8-bit image into a binary image, that consists of the pixels comprising the LDs. Following binarization, the image was subjected to watershed object separation for image processing, which is used to identify borders of adjacent LDs (**Abdolhoseini et al., 2019**). After separation, the binary image was manually compared with the original image for consistency and correct binary conversion. After setting the scale of the image, the amount and individual size of the LDs in the image, displayed by ImageJ as surface area in µm2, were measured. Incomplete LDs located at the edge of the image were excluded. The sum of the areas of all the LDs present in a cell was used as the total lipid content.

Staining of neutral lipid and DNA was performed in hADSCs cultured on 96-wells plates under *PRPF8*-silencing as previously described (**Pettersson et al., 2015**). hADSCs were washed with PBS and fixed with 4 % paraformaldehyde solution (PFA) containing 0.123 mol/L $NaH_2PO_4 \times 2H_2O$, 0.1 mol/L NaOH and 0.03 mol/L glucose for 10 min at RT. Fixed cells were washed with PBS and stained with Bodipy 500/510 (0.2 µg/ml PBS) and Hoechst (2 µg/ml PBS, Molecular probes) for 20 min at RT. After washing with PBS, accumulation of intracellular lipids (Bodipy staining) and cell number (Hoechst staining for nuclei) were quantified with Cell Insight CX5 High-Content Screening (HCS) Platform (Thermo Fisher Scientific) and High-Content Screening (HCS) Studio Cell Analysis Software 2.0. Total lipid content, LD number and size were normalised by cell number.

## Experimental treatments

To evaluate the contribution of the SF3B complex to adipogenesis, we employed pladienolide-B, a natural product that binds to this complex and inhibits pre-mRNA splicing via targeting splicing factor SF3B1 (**Aouida et al., 2016**; **Cretu et al., 2018**). SGBS cells at day 4 of differentiation were treated with pladienolide-B or vehicle for 24 hr (**Jiménez-Vacas et al., 2019**). Pladienolide-B was initially used in the $10^{-11}$ to $10^{-7}$ M range, being $10^{-8}$ M the higher dose that did not compromise cell viability (data not shown).

SGBS preadipocytes at day 4 of differentiation were exposed for 30 h, to different treatments inducing IR as previously described (**Díaz-Ruiz et al., 2015**): the inflammatory cytokine tumor necrosis factor alpha (TNFα, 5 nM), a combination of high glucose (4.5 g/L) and high insulin (100 nmol/L) (HGHI), palmitic acid (sodium salt; 500 µM), or oleic acid (500 µM). Palmitic and oleic acid solutions contained 2 % fatty acid-free BSA. One sample per experiment was used to obtain control responses in the presence of the solvent. After the treatments, control and TNFα-, HGHI-, palmitic, and oleic acid-treated SGBS cells were stimulated with insulin (100 nM) for 15 min. Primary human OM preadipocytes from NG obese individuals (cohort 1) at day 3 of differentiation were also exposed to HGHI conditions (24 h).

In a second series of experiments, the effect of a bile acid known to reduce ER stress, tauroursodeoxycholic acid (TUDCA) (**Uppala et al., 2017**; **Xia et al., 2017**; **Zhang et al., 2018**), was tested in human OM preadipocytes from NG obese individuals and SGBS cells exposed to HGHI conditions as well as in human OM preadipocytes from IR/T2D obese individuals. To this end, cells at day 3–4 of differentiation were exposed to 0.5 mg/mL TUDCA for 14 h before culture in medium alone or in the presence of glucose (4.5 g/L) and insulin (100 nmol/L) (HGHI) for 24–30 h. At the end of the experiments, cells were processed for immunoblotting, RNA isolation, and Oil Red-O staining.

## Confocal immunofluorescence microscopy

Samples were visualized by confocal microscopy under an LSM 5 Exciter confocal microscope fitted with Immersol immersion oil. Depending on the cell depth, 5–8 stacks per channel were collected and projected in a single image. After acquisition, images underwent a deconvolution process with the software package Huygens Professional 2.4.4.

## RNA extraction and real-time PCR (RT-PCR)

Total RNA isolation and purification from human preadipocytes were performed using the RNeasy Kit. RNA from primary preadipocytes and SGBS cells under the different experimental conditions tested was isolated using TRIzol Reagent following the manufacturers' protocols, and subsequently treated with DNase. Quantification of recovered RNA was assessed using NanoDrop2000 spectrophotometer. One µg of RNA was retrotranscribed to cDNA as previously described (**Peinado et al., 2010**)

using with the Revertaid First Strand cDNA Synthesis kit. Transcript levels were quantified by RT-PCR with GoTaq qPCR Master Mix kit. Previously validated specific primers (*Supplementary file 3*) were designed using Primer3 Input 4.1.0 and purchased from Metabion (Steinkirchen, Germany). Primers or encompassing fragments of the areas from the extremes of two exons were designed to ensure the detection of the corresponding transcripts avoiding genomic DNA amplification. The cDNA was amplified with a thermal profile at the following conditions: hot-start activation at 95 °C for 2 min, followed by 40 cycles of denaturation (95 °C for 15 s), then annealing/extension (60 °C for 60 s), and finally, a dissociation cycle (melting curve; 60°C to 95°C, increasing 0.5 °C / 30 s) to verify that only one product was amplified, using the Light-Cycler 96 instrument (Roche; Basilea, Switzerland). The primer concentrations were 500 nM. To allow for variation in the amount of RNA used and the efficiency of the reverse-transcription reaction, all results were normalized by the expression of three house-keeping genes (*ACTB*, *GAPDH*, and *HPRT*) and relative quantification was calculated using the ΔCT formula (*Catalán et al., 2007*). All samples for each experiment were run in the same plate in triplicate and the average values were calculated.

## qPCR dynamic array

A qPCR dynamic array based on microfluidic technology for simultaneous determination of the expression of 45 transcripts in 36 samples (Fluidigm; San Francisco, CA) was employed (*Del Río-Moreno et al., 2019*). Specific primers for human transcripts including components of the major (n = 13) and minor spliceosome (n = 4), associated splicing factors (n = 28), and three housekeeping genes (*ACTB*, *GAPDH*, and *HPRT*) were specifically designed with the Primer3 software and StepO-neTM Real-Time PCR System software v2.3 (Applied Biosystems, Foster City, CA). Preamplification, exonuclease treatment, and qPCR dynamic array based on microfluidic technology were implemented as recently reported (*Gahete et al., 2018*; *Del Río-Moreno et al., 2019*), following manufacturer's instructions using the Biomark System and the Real-Time PCR Analysis Software (Fluidigm). The panel of splicing machinery components was selected on the basis of two main criteria: (1) the relevance of the given spliceosome components in the splicing process (such as the components of the spliceo-some core), and (2) their demonstrated participation in the generation of splicing variants implicated in the pathophysiology of metabolic diseases (as is the case of the 28 splicing factors selected in this study) (*Gahete et al., 2018*; *Del Río-Moreno et al., 2019*).

Briefly, following manufacturer's instructions, 12.5 ng of cDNA of each sample were pre-amplified using 1 µL of all primers mix (500 nM) in a T100 Thermal cycler (BioRad, Hercules, CA), using the following program: (1) 2 min at 95 °C; (2) 15 sec at 94 °C and 4 min at 60 °C (14 cycles). Then, samples were treated with 2 µL of 4 U/µL Exonuclease I solution following the manufacturer's instructions. Next, samples were diluted with 18 µL of TE Buffer, and 2.7 µL were mixed with 3 µL of EvaGreen Supermix and 0.3 µL of cDNA Binding Dye Sample Loading Reagent. Primers were diluted to 5 µM with 2 X Assay Loading Reagent. Control line fluid was charged in the chip and Prime script program was run into the IFC controller MX. Finally, 5 µL of each primer and 5 µL of each sample were pipetted into their respective inlets on the chip and the Load Mix script in the IFC controller software was run. Thereafter, the qPCR was run in the Biomark System following the thermal cycling program: (1) 9 °C for 1 min; (2) 35 cycles of denaturing (95 °C for 5 s) and annealing/extension (60 °C for 20 s); and (3) a last cycle where final PCR products were subjected to graded temperature-dependent dissociation (60°C to 95°C, increasing 1 °C/3 s). Data were processed with Real-Time PCR Analysis Software 3.0.

Finally, to control for variations in the efficiency of the retrotranscription reaction, mRNA copy numbers of the different transcripts analysed were adjusted by normalization factor (NF), calculated with the expression levels of the three housekeeping genes mentioned above using GeNorm 3.3 soft-ware. This selection was based on the stability of these housekeeping genes among the experimental groups to be compared, wherein the expression of these housekeeping genes was not significantly different among groups.

## Quantitative immunoblotting

Human preadipocytes isolated from SC and OM adipose tissue of NG, IR and T2D obese individuals were disrupted in BLC buffer containing 150 mM NaCl, 20 mM Tris-HCl pH 7.4, 1 % Triton-X-100, 1 mM EDTA, and 1 µg/mL anti-protease cocktail CLAP (Chymostatin, Leupeptin, Antipain and Pepstatin A). Protein content was measured by Bradford assay and 30 mg of protein per sample were loaded into

4–20% precast SDS-PAGE gels (Bio-Rad) under denaturing conditions. SGBS cells were homogenized in SDS-DTT buffer containing 62.5 mM Tris-HCL pH 7.6, 2 % SDS, 100 mM DTT, 20 % Glycerol. Protein content was separated by 10 % SDS-PAGE under denaturing conditions. In both cases, samples were transferred to nitrocellulose membranes (Bio-Rad). After Ponceau staining to ensure equal sample loading, membranes were blocked in Tris-buffer saline (TBS) consisting of 25 mM Tris, 150 mM NaCl with 0.05 % Tween 20 (TTBS) containing 5 % non-fat dry milk for 1 hr at RT. Blots were then incubated overnight at 4 °C with the corresponding primary antibody (**Appendix 1—key resource table**). Immunoreactive bands were visualized using horseradish peroxidase (HRP)-conjugated secondary antibodies (**Appendix 1—key resource table**) and the enhanced chemiluminescence ECL Plus detection system. Quantification of band intensities was carried out on digital images of membrane samples provided by LAS4000 gel documentation system (GE Healthcare; Barcelona, Spain) using ImageJ software (1.50b, NIH, Bethesda, MA), and normalized with Ponceau density values as previously described (*Jimenez-Gomez et al., 2013*; *Peinado et al., 2011*). All comparative experiments were performed under identical conditions.

## Proteasome activity assay

The 26 S proteasome (ATP stimulated) and/or 20 S (SDS stimulated) activities were measured in primary human OM preadipocytes from NG, IR, and T2D obese individuals (cohort 1) and SGBS preadipocytes upon exposure to HGHI (*Díaz-Ruiz et al., 2015*; *Bunnell et al., 2008*). Assays for proteasome activity were performed using Suc-LLVY-AFC product, which is proteolyzed by the chymotrypsin-like (ChT-L) active site of the proteasome forming a fluorogenic substrate (AFC) (*Keita et al., 2014*).

Cells were washed with D-PBS and scraped from plates in BLC buffer without DTT, TX-100 or protease inhibitors because these compounds interfere with the assay. The reaction was conducted in 250 µL of activity assay buffer in a 96-well plate containing 50 µg/ml protein lysate in Proteasome buffer (20 mM Tris–HCl, pH 7.8, 1 mM EDTA, 0.5 mM DTT, 5 mM $MgCl_2$, and 50 µM Suc-LLVY-AFC) plus 2 mM ATP or 0.02 % SDS for the determination of the 26 S or 20 S proteasome, respectively. In order to confirm the specificity of the findings for proteasome activity, parallel reactions containing the proteasome inhibitor, MG-132 (25 µM), were run as controls. The reaction mixture was incubated for 3 hr at 37 °C and the fluorescence of the released AMC product was kinetically followed each 10 min in a Flex Station three at an emission wavelength of 355 nm and an excitation wavelength of 460 nm. The background fluorescence values obtained by incubating the lysates with MG132 were subtracted from activity values. Results were analysed with the SoftMax Pro 2.2.1 and the values reported for proteasome peptidase activity are expressed as relative fluorescence arbitrary units/min/mg of total protein.

## Cell viability and cytotoxicity assays

Cell viability of SGBS cell cultures was assessed by the MTT assay. Briefly, cells were incubated for 2 h with 0.1 mg/ml of 3-(4,5-dimetililtiazol-2-ilo)–2,5-difeniltetrazol de bromuro (MTT), dissolved in D-PBS. Washing with D-PBS (1 mL) was followed by the addition of DMSO (1 mL) and gentle shaking for 10 min so that complete dissolution was achieved. Aliquots (200 µL) of the resulting solutions were transferred to 96-well plates and absorbance was recorded at 570 nm using a Flex Station 3. Results were analysed with the SoftMax Pro 2.2.1 and are presented as percentage of the control siRNA values.

Cytotoxicity of PRPF8-silencing in hADSCs was evaluated by Cytotoxicity Detection Kit Plus. The culture medium was aspirated and centrifuged at 3000 rpm for 10 min to obtain a cell free supernatant. Aliquots of media and warm reagent were mixed in a 96-well plate and absorbance was recorded using a microplate reader (Tecan Infinite 2000; Männedorf, Switzerland). Results were analysed with Magellan software 7.2 SP1.

## Measurement of cell proliferation rate

Cell counting in culture flasks was carried out to monitor growth rates of the preadipocytes in culture. At passage 3, cells were detached from the culture flasks and an aliquot was stained with trypan blue and cells were counted in a Neubauer chamber. Cell proliferation was calculated as the total number of live cells per initial seeding number (*Morten et al., 2016*).

## Statistical analysis

For in vitro experiments, at least three replicates were obtained for each condition. Data are presented as mean ± standard error of the mean (S.E.M.) and statistical analysis was performed using GraphPad Prism 7. To determine the normality distribution of the samples, Shapiro-Wilk normality test was used. To determine significance, either unpaired t test for parametric data or Mann Whitney test for non-parametric data were employed for comparisons between two groups. For comparison of more than two groups either one-way ANOVA with Tukey's multiple comparisons test for parametric data or Kruskal-Wallis with Dunn's multiple comparisons test for non-parametric were used. A two-way ANOVA was used to identify significant differences between experimental conditions in the response to insulin or during differentiation. Values were considered significant at $p < 0.05$. Statistical details and significance can be found in the figure legends.

## Acknowledgements

The authors are grateful to the Proteomics Facilities of the SEPBIO/Autonomous University of Barcelona (UAB; Spain) and David Ovelleiro (Proteomics Unit, IMIBIC) for the iTRAQ-LC-MS/MS analysis and the Andalusian Bioinformatics Platform (PAB) centre located at the University of Málaga (Spain) for the Ingenuity Pathway Analysis (IPA) software. We thank Dr Wabitsch (Ulm University, Germany) for sharing SGBS cells. We thank Rafael Serrano-Berzosa (IMIBIC) for his technical help. Funding: Ministerio de Ciencia, Innovación y Universidades/FEDER (BFU2013-44229-R, BFU2016-76711-R, BFU2017-90578-REDT to MMM; RTI2018-093919-B-I00 to SF-V); Consejería de Salud y Bienestar Social/Junta de Andalucía/FEDER (PI-0200/2013 to MMM; PI-0159-2016 to RG-R); Instituto de Salud Carlos III (ISCIII)/FEDER (PIE14/00005 to JL-M and MMM; PI16/00264 to RML; PI17/0153 to JV); Fondo de Investigación Sanitaria/ISCIII/FEDER Miguel Servet tenure-track program (CP10 /00438, CPII16/00008 to SF-V); Research Plan of University of Córdoba (Mod 2.5, 2019 to RG-R); Co-funded by European Regional Development Fund/European Social Fund "Investing in your future"; and Consejería de Economía, Conocimiento, Empresas y Universidad/Junta de Andalucía/FEDER (BIO-0139). CIBEROBN is an initiative of the ISCIII, Spain.

## Additional information

### Funding

| Funder | Grant reference number | Author |
| --- | --- | --- |
| Ministerio de Ciencia, Innovación y Universidades | BFU2013-44229-R | María M Malagón |
| Ministerio de Ciencia, Innovación y Universidades | BFU2016-76711-R | María M Malagón |
| Ministerio de Ciencia, Innovación y Universidades | BFU2017-90578-REDT | María M Malagón |
| Ministerio de Ciencia, Innovación y Universidades | RTI2018-093919-B-I00 | Sonia Fernández-Veledo |
| Junta de Andalucía | PI-0200/2013 | María M Malagón |
| Junta de Andalucía | PI-0159-2016 | Rocío Guzmán-Ruiz |
| Instituto de Salud Carlos III | PIE14/00005 | María M Malagón |
| Instituto de Salud Carlos III | PI16/00264 | Raul M Luque |
| Instituto de Salud Carlos III | PI17/0153 | Joan Vendrell |
| Instituto de Salud Carlos III | CP10/ 00438 | Sonia Fernández-Veledo |
| Instituto de Salud Carlos III | CPII16/00008 | Sonia Fernández-Veledo |
| European Social Fund | BIO-0139 | María M Malagón |

| Funder | Grant reference number | Author |
|---|---|---|

The funders had no role in study design, data collection and interpretation, or the decision to submit the work for publication.

## Author contributions

Julia Sánchez-Ceinos, Conceptualization, Data curation, Formal analysis, Funding acquisition, Investigation, Methodology, Performance of research, data analysis, and manuscript writting, Project administration, Resources, Software, Supervision, Validation, Visualization, Writing – original draft, Writing – review and editing; Rocío Guzmán-Ruiz, Conceptualization, Funding acquisition, Methodology, Research design, Supervision, Writing – review and editing; Oriol Alberto Rangel-Zúñiga, Resources, Selection of patients from cohort 1, Selection of patients from cohort 1, Selection of patients from cohort 1, Selection of patients from cohort 1, Writing – review and editing; Jaime López-Alcalá, Investigation, Performance of research and data analysis, Performance of research and data analysis, Performance of research and data analysis, Writing – review and editing; Elena Moreno-Caño, Data curation, Formal analysis, Investigation, Methodology, Performance of research and data analysis, Performance of research and data analysis, Performance of research and data analysis, Writing – review and editing; Mercedes Del Río-Moreno, Data curation, Formal analysis, Investigation, Methodology, Performance of research and data analysis, Performance of research and data analysis, Performance of research and data analysis, Writing – review and editing; Juan Luis Romero-Cabrera, Pablo Pérez-Martínez, Resources, Selection of patients from cohort 1, Selection of patients from cohort 1, Selection of patients from cohort 1, Selection of patients from cohort 1, Writing – review and editing; Elsa Maymo-Masip, Funding acquisition, Investigation, Methodology, Resources, Selection of patients from cohort 2 and performance of preadipocyte isolation, Selection of patients from cohort 2 and performance of preadipocyte isolation, Selection of patients from cohort 2 and performance of preadipocyte isolation, Writing – review and editing; Joan Vendrell, Funding acquisition, Resources, Selection of patients from cohort 2 and performance of preadipocyte isolation, Selection of patients from cohort 2 and performance of preadipocyte isolation, Selection of patients from cohort 2 and performance of preadipocyte isolation, Writing – review and editing; Sonia Fernández-Veledo, Funding acquisition, Investigation, Resources, Selection of patients from cohort 2 and performance of preadipocyte isolation, Selection of patients from cohort 2 and performance of preadipocyte isolation, Selection of patients from cohort 2 and performance of preadipocyte isolation, Writing – review and editing; José Manuel Fernández-Real, Investigation, Participation in SGBS experiments, Resources, Writing – review and editing; Jurga Laurencikiene, Mikael Rydén, Investigation, Participated in hADSCs experiments, Participated in hADSCs experiments, Resources, Writing – review and editing; Antonio Membrives, Resources, Selection patients from cohort 1, Writing – review and editing; Raul M Luque, Conceptualization, Coordination of the splicing array study, Funding acquisition, Investigation, Methodology, Resources, Writing – review and editing; José López-Miranda, Funding acquisition, Investigation, Resources, Selection of patients from cohort 1, Selection of patients from cohort 1, Selection of patients from cohort 1, Selection of patients from cohort 1, Writing – review and editing; María M Malagón, Conceptualization, Funding acquisition, Investigation, Project administration, Research design, Research design, Resources, Supervision, Writing – original draft, Writing – review and editing

## Author ORCIDs

Julia Sánchez-Ceinos (ID) http://orcid.org/0000-0002-3131-7057
María M Malagón (ID) http://orcid.org/0000-0002-2419-2727

## Ethics

Human subjects: Written consent was obtained from all the participants prior to recruitment, and the experimental protocol was approved by the Ethics and Research Committee of corresponding hospitals following the Helsinki Declaration (ref 3170).

## Decision letter and Author response

Decision letter https://doi.org/10.7554/eLife.65996.sa1
Author response https://doi.org/10.7554/eLife.65996.sa2

## Additional files

### Supplementary files

• Supplementary file 1. Lineal regression analyses of subcutaneous (SC) and omental (OM) preadipocytes proliferation rate and anthropometric and biochemical parameters, = 24–58. $r$ = Spearman's correlation coefficient. $*P < 0.05$, $**P < 0.01$, $***P < 0.001$. Normality distribution was determined by Shapiro-Wilk normality test.

• Supplementary file 2. Anthropometric and biochemical characteristics of study subjects from cohort 2. NG, normoglycemic; T2D, type 2 diabetes; LDL, low-density lipoprotein; HDL, high-density lipoprotein; HOMA-IR, homeostasis model assessment of insulin resistance. [aa]P <0.01, [aaa]P <0.001 *vs.* Lean; [b]P <0.05 *vs.* NG Obese. One-way ANOVA with Tukey's multiple comparisons test or Kruskal-Wallis with Dunn's multiple comparisons test (for parametric or non-parametric data, respectively) were used. Normality distribution was determined by Shapiro-Wilk normality test.

• Supplementary file 3. Sequences and transcript sizes of primers used for RT-PCR studies. bp, base pairs.

• Transparent reporting form

### Data availability

All the proteomic data related to this study are available at the ProteomeXchange Consortium via the Proteomics IDEntifications (PRIDE) partner repository with the dataset identifier PXD015621. All data generated or analysed during this study are included in the manuscript and supporting files. Source data files have been provided for Figures 1 and 3.

The following dataset was generated:

| Author(s) | Year | Dataset title | Dataset URL | Database and Identifier |
|---|---|---|---|---|
| Sánchez-Ceinos J, Guzmán-Ruiz R, Rangel-Zuñiga OA, Moreno-Caño E, del Río-Moreno M, Maymo-Masip E, Vendrell J, Fernández-Veledo S, Laurencikiene J, Rydén M, Membrives A, Luque RM, López-Miranda J, Malagón MM | 2021 | Obese preadipocyte proteome | http://www.proteomexchange.org/ | ProteomeXchange Team, PXD015621 |

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

# Appendix 1

## Appendix 1—key resources table

| Reagent type (species) or resource | Designation | Source or reference | Identifiers | Additional information |
|---|---|---|---|---|
| Cell line (*Homo sapiens*) | Human preadipocytes from SC and OM adipose tissue | This paper | N/A | Primary cell line |
| Cell line (*Homo sapiens*) | Human adipose-derived stromal cells (hADSCs) | This paper and previous work from Dr. Mikael Rydén group *Gao et al., 2014*; *Gao et al., 2017* | | SC adipose tissue from a male donor (16 years old, BMI 24 kg/m²) |
| Cell line (*Homo sapiens*) | Simpson-Golabi-Behmel syndrome (SGBS) cell line | Gift from Prof. Dr. José Manuel Fernández-Real (Institut d'Investigació Biomèdica; Girona, Spain) | N/A | SC adipose tissue from a 3 months male infant with SGBS |
| Biological sample (*Homo sapiens*) | Human SC and OM adipose tissue from NG, IR and T2D morbid obese individuals (cohort 1) | General and Digestive Surgery Unit and the Lipids and Atherosclerosis Unit of the Reina Sofía University Hospital (Córdoba, Spain) | Reina Sofía University Hospital Research Ethical Committee (439/2010) | Ethics Committee HURS, ref 3170 |
| Biological sample (*Homo sapiens*) | Human SC and OM adipose tissue from lean, NG and, T2D obese individuals (cohort 2) | Endocrinology ad Surgery Departments at the University Hospital Joan XXIII (Tarragona, Spain) *Ejarque et al., 2017*; *Serena et al., 2016* | | N/A |
| Antibody | ADRP (B-6) (mouse monoclonal) | Santa Cruz Biotechnology | sc-377429, N/A | (1:1000) |
| Antibody | AKT (rabbit polyclonal) | Cell Signaling Technology | 9272, AB_329827 | (1:1000) |
| Antibody | ATF6α (H-280) (rabbit polyclonal) | Santa Cruz Biotechnology | sc-22799, AB_2242950 | (1:1000) |
| Antibody | CASPASE-3 (8G10) (rabbit monoclonal) | Cell Signaling Technology | 9665, AB_2069872 | (1:1000) |
| Antibody | CD14 (M-305) (rabbit polyclonal) | Santa Cruz Biotechnology | sc-9150, AB_2074171 | (1:1000) |
| Antibody | CD45 (H-230) (rabbit polyclonal) | Santa Cruz Biotechnology | sc-25590, AB_2174143 | (1:1000) |
| Antibody | CHOP (L63F7) (mouse monoclonal) | Cell Signaling Technology | 2895, AB_2089254 | (1:1000) |
| Antibody | DGAT2 (goat polyclonal) | Novus | NB100-57851, AB_921135 | (1:1000) |
| Antibody | DLK/PREF1 (H-118) (rabbit polyclonal) | Santa Cruz Biotechnology | sc-25437, AB_2292943 | (1:1000) |
| Antibody | eIF2α (3A7A8) (mouse monoclonal) | Santa Cruz Biotechnology | sc-517214, N/A | (1:1000) |
| Antibody | FAS (C-20) (goat polyclonal) | Santa Cruz Biotechnology | sc-16147, AB_2101097 | (1:1000) |
| Antibody | BiP/GRP78 (A-10) (mouse monoclonal) | Santa Cruz Biotechnology | sc-376768, N/A | (1:1000) |
| Antibody | GRP94 (rabbit polyclonal) | Cell Signaling Technology | 2104, AB_823506 | (1:1000) |
| Antibody | HSL/LIPE (rabbit polyclonal) | Abcam | ab45422, AB_2135367 | (1:1000) |
| Antibody | IRE1α (B-12) (mouse monoclonal) | Santa Cruz Biotechnology | sc-390960, N/A | (1:1000) |
| Antibody | PDI (C81H6) (rabbit monoclonal) | Cell Signaling Technology | 3501, AB_2156433 | (1:1000) |
| Antibody | PERK (B-5) (mouse monoclonal) | Santa Cruz Biotechnology | sc-377400, AB_2762850 | (1:1000) |

*Appendix 1 Continued on next page*

*Appendix 1 Continued*

| Reagent type (species) or resource | Designation | Source or reference | Identifiers | Additional information |
|---|---|---|---|---|
| Antibody | Phospho-AKT (Ser473) (rabbit monoclonal) | Cell Signaling Technology | 4060, AB_2315049 | (1:750) |
| Antibody | Phospho-eIF2α (Ser51) (rabbit polyclonal) | Santa Cruz Biotechnology | sc-101670, AB_2096507 | (1:750) |
| Antibody | Phospho-HSL/LIPE (Ser563) (rabbit polyclonal) | Cell Signaling Technology | 4139, AB_2135495 | (1:750) |
| Antibody | Phospho-IRE1α (Ser724) (rabbit polyclonal) | Abcam | ab48187, AB_873899 | (1:750) |
| Antibody | Phospho-PERK (Thr981) (rabbit polyclonal) | Santa Cruz Biotechnology | sc-32577, AB_2293243 | (1:750) |
| Antibody | PLIN1 (guinea pig polyclonal) | Progen | GP29, AB_2892611 | (1:1000) |
| Antibody | PPARγ (D69) (rabbit polyclonal) | Cell Signaling Technology | 2430, AB_823599 | (1:1000) |
| Antibody | PRP8 (E-5) (mouse monoclonal) | Santa Cruz Biotechnology | sc-55533, AB_831685 | (1:1000) |
| Antibody | PSMB8/LMP7 (sheep polyclonal) | R&D Systems | AF7710, N/A | (1:1000) |
| Antibody | SFPQ (rabbit monoclonal) | Abcam | ab177149, N/A | (1:1000) |
| Antibody | UBIQUITIN (rabbit polyclonal) | Cell Signaling Technology | 3933, AB_2180538 | (1:1000) |
| Antibody | Goat/Sheep IgG (mouse monoclonal) | Sigma-Aldrich | A9452, AB_258449 | (1:2500) |
| Antibody | Guinea Pig IgG (goat polyclonal) | Sigma-Aldrich | A7289, AB_258337 | (1:2500) |
| Antibody | Mouse IgG (rabbit polyclonal) | Sigma-Aldrich | A9044, AB_258431 | (1:2500) |
| Antibody | Rabbit IgG (goat polyclonal) | Sigma-Aldrich | A8275, AB_258382 | (1:10000) |
| Recombinant DNA reagent | phrGFP-N1 | Agilent Technologies | 240,036 | |
| Recombinant DNA reagent | pcDNA3.1(+) | Invitrogen | V79020 | |
| Recombinant DNA reagent | *PRPF8* (*NM_006445*) in pcDNA3.1(+) | GenScript | OHu19527C | |
| Recombinant DNA reagent | *SFPQ* (*NM_005066.3*) in pcDNA3.1(+) | GenScript | OHu23607C | |
| Recombinant DNA reagent | pCMV-Myc | Clontech | 635,689 | |
| Recombinant DNA reagent | pCMV-BiP-Myc-KDEL-wt | AddGene | 27,164 Addgene_27164 | |
| Sequence-based reagent | Primers for RT-PCR, see *Supplementary file 3* | This paper | N/A | |
| Sequence-based reagent | Primers for splicing-machinery components array | *Gahete et al., 2018* | N/A | |
| Sequence-based reagent | *PRPF8* Silencer Select Validated siRNA | Ambion | 4390824 | |
| Sequence-based reagent | *SFPQ* Silencer Select Validated siRNA | Ambion | s224606 | |
| Sequence-based reagent | Silencer Select Negative Control No. 1 siRNA | Ambion | 4390843 | |
| Peptide, recombinant protein | 3-Isobutyl-1-methylxanthine (IBMX) | Sigma-Aldrich | I5879 | |

*Appendix 1 Continued on next page*

*Appendix 1 Continued*

| Reagent type (species) or resource | Designation | Source or reference | Identifiers | Additional information |
|---|---|---|---|---|
| Peptide, recombinant protein | Antipain | Sigma-Aldrich | A6191 | |
| Peptide, recombinant protein | Biotin | Sigma-Aldrich | B4639 | |
| Peptide, recombinant protein | Bovine Serum Albumin (BSA) | Sigma-Aldrich | A8806, A7030 | |
| Peptide, recombinant protein | Chymostatin | Sigma-Aldrich | C7268 | |
| Peptide, recombinant protein | Collagenase Type V | Sigma-Aldrich | C9263 | |
| Peptide, recombinant protein | Dexamethasone | Sigma-Aldrich | D4902, D1756 | |
| Peptide, recombinant protein | Human 3,3′,5-Trihydrochloride sodium salt (T3) | Sigma-Aldrich | T5516, T6397 | |
| Peptide, recombinant protein | Human FGF2 | Sigma-Aldrich | F0291 | |
| Peptide, recombinant protein | Human Insulin | Sigma-Aldrich | I2643, I9278 | |
| Peptide, recombinant protein | Human Transferrin | Sigma-Aldrich | T8158 | |
| Peptide, recombinant protein | Human Tumor Necrosis Factor-$\alpha$ (TNF$\alpha$) | Sigma-Aldrich | T6674 | |
| Peptide, recombinant protein | Hydrocortisone | Sigma-Aldrich | H0135 | |
| Peptide, recombinant protein | Leupeptin | Sigma-Aldrich | L2884 | |
| Peptide, recombinant protein | MG-132 | Calbiochem | 474,790 | |
| Peptide, recombinant protein | N-Succinyl-Leu-Leu-Val-Tyr-7-Amido-4-Methylcoumarin (Suc-LLVY-AFC) | Sigma-Aldrich | S6510 | |
| Peptide, recombinant protein | Pepstatin A | Sigma-Aldrich | P4265 | |
| Peptide, recombinant protein | Rosiglitazone | Sigma-Aldrich | R2408 | |
| Chemical compound, drug | (+)-Sodium L-ascorbate | Sigma-Aldrich | A4034 | |
| Chemical compound, drug | 1,4-dithiothreitol (DTT) | Thermo Scientific | R0862 | |
| Chemical compound, drug | Acetonitrile | Sigma-Aldrich | 271,004 | |
| Chemical compound, drug | Adenosine 5′-triphosphate (ATP) disodium salt hydrate | Sigma-Aldrich | A26209 | |

*Appendix 1 Continued on next page*

*Appendix 1 Continued*

| Reagent type (species) or resource | Designation | Source or reference | Identifiers | Additional information |
|---|---|---|---|---|
| Chemical compound, drug | BODIPY 500/510 $C_1$, $C_{12}$ (4,4-Difluoro-5-Methyl-4-Bora-3a,4a-Diaza-s-Indacene-3-Dodecanoic Acid) | Invitrogen | D3823 | |
| Chemical compound, drug | Calcium chloride ($CaCl_2$) | Sigma-Aldrich | 449,709 | |
| Chemical compound, drug | CHAPS hydrate | Sigma-Aldrich | C3023 | |
| Chemical compound, drug | Chloroform | Sigma-Aldrich | C2432 | |
| Chemical compound, drug | Clarity Western ECL Substrate | Bio-Rad | 1705061 | |
| Chemical compound, drug | D-(+)-Glucose | Panreac Applichem / Siga-Aldrich | 1413411211, G8270 | |
| Chemical compound, drug | DAPI | Sigma-Aldrich | D9542 | |
| Chemical compound, drug | Dimethyl sulfoxide (DMSO) | Sigma-Aldrich | D4540 | |
| Chemical compound, drug | D-Pantothenic acid hemicalcium salt | Sigma-Aldrich | P5155 | |
| Chemical compound, drug | Dulbecco's Phosphate Buffered Saline (D-PBS) Solution | Sigma-Aldrich | D1408 | |
| Chemical compound, drug | Dulbecco's Phosphate Buffered Saline (D-PBS) Solution (-) $Ca^{2+}$, $Mg^{2+}$ | HyClone | SH30028.02 | |
| Chemical compound, drug | Dulbecco's Modified Eagle Medium (DMEM)/F-12 (1:1) | Gibco | 31330–038 | |
| Chemical compound, drug | EDTA disodium salt | Sigma-Aldrich | E5134 | |
| Chemical compound, drug | Exonuclease I Reaction Buffer | New England BioLabs | B0293S | |
| Chemical compound, drug | Fluorescence Mounting Medium | Dako | S3023 | |
| Chemical compound, drug | Formic acid | Scharlau | AC10760050 | |
| Chemical compound, drug | Glycerol | VWR | 97063–892 | |
| Chemical compound, drug | Ham's F-12 Nutrient Mix | Gibco | 21765037 | |
| Chemical compound, drug | Hepes Buffer (1 M) | Gibco | 15630056 | |
| Chemical compound, drug | Hoechst 33342, Trihydrochloride, Trihydrate | Invitrogen | H3570 | |
| Chemical compound, drug | Immersol Immersion Oil | Carl Zeiss | 518 F | |
| Chemical compound, drug | Iodoacetamide (IAA) | Sigma-Aldrich | I1149 | |
| Chemical compound, drug | Isopropanol | Sigma-Aldrich | 33,539 | |
| Chemical compound, drug | iTRAQ Reagents | Applied Biosystems | PN4351918 | |
| Chemical compound, drug | Lipofectamine 2000 Transfection Reagent | Invitrogen | 11668019 | |
| Chemical compound, drug | Lipofectamine RNAiMAX Transfection Reagent | Invitrogen | 13778150 | |

*Appendix 1 Continued on next page*

*Appendix 1 Continued*

| Reagent type (species) or resource | Designation | Source or reference | Identifiers | Additional information |
|---|---|---|---|---|
| Chemical compound, drug | Magnesium Chloride (MgCl$_2$) | Merck | 5,833 | |
| Chemical compound, drug | Magnesium Sulfate 7-hydrate (MgSO$_4$) | Panreac Applichem | 141,404 | |
| Chemical compound, drug | Methanol (MeOH) | VWR Chemicals | 20864320 | |
| Chemical compound, drug | Newborn Calf Serum (NCS) | Gibco | 16010159 | |
| Chemical compound, drug | Nonfat dried milk powder | Panreac Applichem | Cat# A0830 | |
| Chemical compound, drug | Oil Red O | Sigma-Aldrich | O0625 | |
| Chemical compound, drug | Oleic acid | Sigma-Aldrich | O1383 | |
| Chemical compound, drug | Opti-MEM I Reduced Serum Medium | Gibco | 11058021 | |
| Chemical compound, drug | Paraformaldehyde (PFA) | Bosterbio | AR1068 | |
| Chemical compound, drug | Penicillin-Streptomycin (10,000 U/mL) | Sigma-Aldrich / Gibco | P4333, 15140122 | |
| Chemical compound, drug | Pladienolide-B | Santa Cruz Biotechnology | sc-391691 | |
| Chemical compound, drug | Potassium Chloride (KCl) | Merck | 104,936 | |
| Chemical compound, drug | Ponceau S | Sigma-Aldrich | P3504 | |
| Chemical compound, drug | RBC Lysis Buffer | Norgen Biotek Corp. | 21,201 | |
| Chemical compound, drug | RNase-Free DNase Set | Qiagen | 79,254 | |
| Chemical compound, drug | Sodium Chloride (NaCl) | Merk Millipore | 7647145 | |
| Chemical compound, drug | Sodium Dodecyl Sulfate (SDS) | Sigma-Aldrich | L3771 | |
| Chemical compound, drug | Sodium Hydroxide (NaOH) | Sigma-Aldrich | S4085 | |
| Chemical compound, drug | Sodium Palmitate | Sigma-Aldrich | P9767 | |
| Chemical compound, drug | Sodium Phosphate Monobasic Monohydrate (NaH$_2$PO$_4$) | Fisher scientific | BP330-1 | |
| Chemical compound, drug | Sodium Phosphate Monobasic Monohydrate Dihydrate (NaH$_2$PO$_4 \times 2H_2O$) | Sigma-Aldrich | 71,505 | |
| Chemical compound, drug | TE Buffer | Invitrogen | 12090015 | |
| Chemical compound, drug | Tetraethylammonium Bromide (TEAB) | Sigma-Aldrich | 241,059 | |
| Chemical compound, drug | Thiazolyl Blue Tetrazolium Bromide (MTT) | Sigma-Aldrich | M5655 | |
| Chemical compound, drug | Trichloroacetic acid (TCA) | Sigma-Aldrich | T6399 | |
| Chemical compound, drug | Triton X-100 | Sigma-Aldrich | T8787 | |
| Chemical compound, drug | Trizma base (Tris base) | Sigma-Aldrich | T6066 | |

*Appendix 1 Continued on next page*

*Appendix 1 Continued*

| Reagent type (species) or resource | Designation | Source or reference | Identifiers | Additional information |
|---|---|---|---|---|
| Chemical compound, drug | Trizma hydrochloride (Tris-HCl) | Sigma-Aldrich | T3253 | |
| Chemical compound, drug | TRIzol Reagent | Invitrogen | 15596018 | |
| Chemical compound, drug | Trypan Blue solution | Sigma-Aldrich | T8154 | |
| Chemical compound, drug | Trypsin Gold, Mass Spectrometry Grade | Promega | V5280 | |
| Chemical compound, drug | Trypsin-EDTA solution | Sigma-Aldrich | T3924 | |
| Chemical compound, drug | Tween 20 | Panreac Applichem | A1389 | |
| Chemical compound, drug | Urea | Sigma-Aldrich | U1205 | |
| Commercial assay or kit | 96.96 DNA Binding Dye Sample / Loading Kit | Fluidigm | BMK-M10-96.96-EG | |
| Commercial assay or kit | AllPrep DNA/RNA/Protein Mini Kit | Qiagen | 80,004 | |
| Commercial assay or kit | Cytotoxicity Detection Kit Plus | Roche | 4744926001 | |
| Commercial assay or kit | GoTaq qPCR Master Mix | Promega | A6001 | |
| Commercial assay or kit | Mycoplasma gel detection kit | Biotools | 4,542 | |
| Commercial assay or kit | NEON Transfection System Kit | Invitrogen | MPK10096 | |
| Commercial assay or kit | Protein Assay Dye Reagent Concentrate | Bio-Rad | 5000006 | |
| Commercial assay or kit | RC DC Protein Assay | Bio-Rad | 5000122 | |
| Commercial assay or kit | RevertAid First Strand cDNA Synthesis Kit | Thermo Scientific | K1621 | |
| Commercial assay or kit | SsoFast EvaGreen Supermix | Bio-Rad | 1725200 | |
| Software, algorithm | Biomark & EP1 Software 4.5.2 | Fluidigm SCR_015685 | https://www.fluidigm.com/software | |
| Software, algorithm | Basic Local Alignment Search Tool (BLAST) | NCBI **Nowicki et al., 2018** SCR_004870 | https://blast.ncbi.nlm.nih.gov/Blast.cgi | |
| Software, algorithm | ENCORI: The Encyclopedia of RNA Interactomes | **Li et al., 2014** N/A | http://starbase.sysu.edu.cn/index.php | |
| Software, algorithm | GeNorm 3.3 | **Vandesompele et al., 2002** SCR_006763 | https://genorm.cmgg.be/ | |
| Software, algorithm | GraphPad PRISM 7 | GraphPad Software, Inc **Mitteer et al., 2018** SCR_002798 | https://www.graphpad.com | |
| Software, algorithm | High-Content Screening (HCS) Studio Cell Analysis Software 2.0 | Thermo Fisher Scientific SCR_018706 | https://www.thermofisher.com/es/es/home/life-science/cell-analysis/cellular-imaging/high-content-screening/hcs-studio-2.html | |
| Software, algorithm | HumanBase | **Greene et al., 2015** SCR_016145 | https://hb.flatironinstitute.org/ | |
| Software, algorithm | Huygens Professional 2.4.4 | Scientific Volume Imaging SCR_014237 | https://svi.nl/Huygens-Professional | |

*Appendix 1 Continued on next page*

*Appendix 1 Continued*

| Reagent type (species) or resource | Designation | Source or reference | Identifiers | Additional information |
|---|---|---|---|---|
| Software, algorithm | ImageJ 1.50b | National Institute of Health (NIH) *Schneider et al., 2012* SCR_001935 | https://imagej.nih.gov/ij/ | |
| Software, algorithm | Ingenuity Pathways Analysis (IPA) 49309495 | Qiagen, provided by the Andalusian Bioinformatics Platform (PAB) center (University of Málaga, Spain; http://www.scbi.uma.es) *Krämer et al., 2014* SCR_008653 | http://www.ingenuity.com/ | |
| Software, algorithm | LightCycler 96 1.1.0.1320 | Roche Life Science SCR_012155 | https://lifescience.roche.com /en_es/products/light cycler-381711.html | |
| Software, algorithm | Magellansoftware 7.2 SP1 | Tecan SCR_008715 | https://lifesciences.tecan.com /software-magellan | |
| Software, algorithm | MetaboAnalyst 4.0 | *Chong et al., 2018* SCR_015539 | https://www.metaboanalyst. ca/docs/About.xhtml | |
| Software, algorithm | Primer3 Input 4.1.0 | *Untergasser et al., 2012* SCR_003139 | http://bioinfo.ut.ee/primer3/ | |
| Software, algorithm | Protein ANalysis THrough Evolutionary Relationships (PANTHER) classification system 14.1 | *Mi et al., 2019* SCR_004869 | http://www.pantherdb.org/ | |
| Software, algorithm | Real-Time PCR Analysis Software 3.0 | Fluidigm SCR_015686 | https://www.fluidigm.com/ | |
| Software, algorithm | SoftMax Pro 2.2.1 | Molecular Devices SCR_014240 | https://www.molecular devices.com/ | |
| Software, algorithm | SpliceAid-F | *Giulietti et al., 2013* SCR_002082 | http://srv00.recas.ba .infn.it/SpliceAidF/ | |
| Software, algorithm | Statistic R module | Pearl software SCR_002394 | https://www.pearlsoftware.com/ | |
| Software, algorithm | StepOne Real-Time PCR System 2.3 | Thermo Scientific SCR_014281 | https://www.thermofisher. com/es/es/home.html | |
| Software, algorithm | Stratagene Mx3000p | Thermo Scientific SCR_020526 | https://www.thermofisher .com/es/es/home.html | |
| Software, algorithm | Venn Diagram Plotter | Pacific Northwest National Laboratory SCR_012842 | https://omics.pnl.gov/software/ venn-diagram-plotter | |

