## [Decision Letter]

**Acceptance summary:**

This manuscript was designed to examine the cause of functional impairment of hypertrophied pre-adipocytes in humans with morbidly obese individuals with metabolic disease. Decreased expression of the core splicing factor PRP8 and also reduced proteasome activity are reported to contribute to the functional defects in pre-adipocytes. The study provides important new insight into mechanisms of metabolic disease, including diabetes.

**Decision letter after peer review:**

Thank you for submitting your article "Impaired mRNA splicing and proteostasis in preadipocytes in obesity-related metabolic disease" for consideration by *eLife*. Your article has been reviewed by 2 peer reviewers, and the evaluation has been overseen by a Reviewing Editor and David James as the Senior Editor. The reviewers have opted to remain anonymous.

Essential revisions:

1 – The authors conclude that decreased PRP8 expression could explain functional impairment of hypertrophied preadipocytes in obesity. However, this appears to be based on correlation and seems to be specific of SC but not OM preadipocytes (Figure 2C) without a clear explanation. In addition, SC preadipocytes from lean subjects have a PRP8 expression comparable to T2D obese group (Figure 2E). It is unclear if the changes in PRP8 expression have functional consequences in these cells or if these changes are a cause or a consequence of impairment of SC preadipocytes. To clarify this point, the authors should investigate if PRP8 complementation is associated with normalized proliferation and/or differentiation potential of SC preadipocytes from obese TD2 patients.

It should also be noted that the acute inactivation of core splicing components (e.g PRP8) in SGBS cell line is expected to have a broad impact in gene expression and adipocyte function. To demonstrate that changes in PRP8 expression have a specific impact in preadipocyte fitness, authors should clarify if changes in the expression of PPARG or SREBF1 isoforms are specifically recapitulated in SC preadipocytes from obese TD2 patients. In addition, they should interrogate whether these changes are specifically normalized by recovering PRP8 expression.

2 – While the study focused on comparing the molecular alterations between obese NG and obese IR/T2D, the levels of the genes/proteins of interest in lean subjects need to be considered. In particular, Figure 2E showed the expression levels of PRPF8 in lean subjects were significantly lower than that in the NG obese people. This would impose a critical caveat for the siRNA silencing studies because reduced PRPF8 level could represent the lean SC preadipocyte. Please discuss the overall implications of the findings when data from lean subjects are considered.

3 – The authors report increased expression of components of the ERAD pathway in preadipocytes from obese TD2 patients. While this doesn't seem to be associated with a consistent increase in UPR response, the authors conclude that in cells obtained from this subgroup of patients there is an impairment in proteostasis. Further mechanistic studies have been performed in a completely independent model (SGBS cell lines) but have not been replicated in preadipocytes obtained from patients (that are the subject of this study). To validate the contribution of ERAD and proteolysis to the functional impairment of preadipocytes in obese patients, these experiments should be performed in preadipocytes obtained from patients. Is proteasome activity decreased in preadipocytes from obese TD2 patients? Is it normalized by chemical chaperones? Is proliferation and/or differentiation potential of SC preadipocytes from obese TD2 patients recovered by chemical chaperones? Does Bip play a central role in orchestrating this response in these cells?

4 – The rigor of data presentation: For the data presented, please show each individual value using the dot plot to let the readers see the data's distributions and ranges of variations. For in vitro studies, please indicate the numbers of technical replicates within a given experiment and the numbers of replicate studies. Please elaborate on how Western blots' data were quantified, including imaging quantification and normalization of loading.

5 – The authors report two mechanisms linked to dysfunction of adipocyte differentiation and function in obese subjects with IR or T2D (i.e., the spliceosome and the ERAD). Can the authors discuss and clarify whether each of these pathways plays an essential, complementary, or redundant role in the pathologies of adipocytes dysfunction in IR or T2D subjects? It would be informative to include an integrative model to present the two pathways and how they regulate adipocyte differentiation and function.

---

## [Author Response]

Essential revisions:1 – The authors conclude that decreased PRP8 expression could explain functional impairment of hypertrophied preadipocytes in obesity. However, this appears to be based on correlation and seems to be specific of SC but not OM preadipocytes (Figure 2C) without a clear explanation. In addition, SC preadipocytes from lean subjects have a PRP8 expression comparable to T2D obese group (Figure 2E). It is unclear if the changes in PRP8 expression have functional consequences in these cells or if these changes are a cause or a consequence of impairment of SC preadipocytes. To clarify this point, the authors should investigate if PRP8 complementation is associated with normalized proliferation and/or differentiation potential of SC preadipocytes from obese TD2 patients.It should also be noted that the acute inactivation of core splicing components (e.g PRP8) in SGBS cell line is expected to have a broad impact in gene expression and adipocyte function. To demonstrate that changes in PRP8 expression have a specific impact in preadipocyte fitness, authors should clarify if changes in the expression of PPARG or SREBF1 isoforms are specifically recapitulated in SC preadipocytes from obese TD2 patients. In addition, they should interrogate whether these changes are specifically normalized by recovering PRP8 expression.

We concur with the Editor and Reviewers in that our proposal that decreased *PRPF8*/PRP8 expression could explain functional impairment of hypertrophied SC preadipocytes in morbidly obese individuals with metabolic disease (IR/T2D), as compared to their normoglycemic counterparts (Figures 2B and 2C in the original version of the manuscript), was based on correlation analyses. In fact, this prevented us from discussing whether *PRPF8*/PRP8 down-regulation was a cause or a consequence of preadipocyte dysfunction. Likewise, we were cautious when discussing the differences found between SC and OM preadipocytes and only suggested that inter-depot differences appeared to be cell autonomous (page 18, lines 429-432, in the original version of the manuscript). In this regard, we proposed that experiments in human primary preadipocytes with altered expression levels of selected splicing-related genes would be required to establish the impact of splicing on adipogenesis (page 18, lines 432-435, in the original version of the manuscript).

With this in mind, and given the comments offered by the Reviewing Editor, we have carried out new experiments using human preadipocytes obtained from obese subjects with T2D. To be more specific, SC preadipocytes from obese patients with T2D were transfected using a *PRPF8*-pcDNA3.1 expression vector to increase *PRPF8* content, as it is shown in the new Figure 4A that has been prepared for the revised version of the manuscript. To complement these data, we employed an identical strategy using SC preadipocytes from obese patients with IR. After transfection, both T2D and IR obese preadipocytes exhibited *PRPF8* expression levels comparable to those of SC preadipocytes from obese NG patients (new Figure 4A). *PRPF8* expression recovery upon *PRPF8*-pcDNA3.1 transfection increased the expression levels of several PPARG isoforms (*PPARG-2*, *PPARG-3*, *PPARG-4*, and γ*ORF4*) to control levels (i.e., SC preadipocytes in NG obesity), especially in SC preadipocytes from obese individuals with IR (new Figure 4C)*.* Similar results were observed for both total *SREBF1* and *SREBF1* isoform 1 (*SREBF1*-*1*) (Figure 4D). In all, these results indicate that the expression of *PPARG* and *SREBF1* could be recapitulated in SC preadipocytes from obese IR/T2D patients by *PRPF8* complementation. In line with these findings are our data from confocal microscopy demonstrating that *PRPF8* complementation by *PRPF8*-pcDNA3.1 expression increased total lipid content in SC preadipocytes from obese subjects with IR/T2D (new Figure 4B). To be more specific, our data show that *PRPF8*-pcDNA3.1 expression increased total lipid content in SC preadipocytes from obese subjects with IR/T2D by increasing lipid droplet size, while decreasing lipid droplet number (new Figure 4B).

Along with these studies, we also examined the effect of *PRPF8* silencing in SC preadipocytes from obese NG patients using a specific siRNA as a mean to mimic the down-regulation of this gene found in SC preadipocytes from obese subjects with IR/T2D. As shown in new Figures 4C and 4D, *PRPF8* down-regulation caused numerical decreases, that in most cases reached statistical significance, in the expression levels of both total *PPARG* and *SREBF1* and their isoforms as compared to mock-transfected SC preadipocytes from NG obese subjects. Notably, rescue experiments by transfection of silenced NG obese preadipocytes with the *PRPF8*-pcDNA3.1 expression vector showed restored expression of most of the genes tested (new Figures 4C and 4D). In accordance with these findings, SC preadipocytes from NG obese patients exhibited lower lipid content and changes in lipid droplet size and number when silenced for *PRPF8*, while these effects were restored upon *PRPF8* complementation (new Figure 4B).

Analysis of markers of lipid droplet biogenesis and growth (*BSCL^-^2*, *CIDEB*, *CIDEC*), showed similar trends to those observed for *PPARG* and *SREBF1* upon manipulation of *PRPF8* expression in SC preadipocytes from either NG or IR/T2D obese individuals (new Figures 4E-4G).

When viewed together, these results highlight the relevance of PRP8 in adipocyte differentiation and further support our initial proposal that impaired PRPF8 expression likely contributes to preadipocyte dysfunction and altered adipogenesis in SC fat in obese individuals with IR/T2D. These new results have been described in a new Results subsection “Analysis of PRPF8/PRP8 effects on human primary preadipocytes” in pages 13 and 14, and discussed in pages 21 and 22 in the new version of the manuscript.

According to our original data from qPCR array and RT-PCR studies of preadipocytes from the three groups of obese individuals (cohort 1; Figure 2—Figure supplements 1 and 2, respectively), not only *PRPF8* but other spliceosome components and splicing factors were decreased in SC preadipocytes from IR/T2D *vs*. NG obese individuals, suggesting that they could be also important for adipocyte differentiation. In fact, recovering *PRPF8*/PRP8 expression in SC preadipocytes from obese individuals with IR/T2D normalized most but not all the changes observed in these cells as compared to their NG counterparts (new Figure 4). To test this hypothesis, and in line with the comment offered by the Reviewing Editor as to whether the changes are normalized by recovering *PRPF8* expression, we carried out additional experiments targeting another component of the major spliceosome, the SF3B complex, and a splicing factor, SFPQ. These experiments were carried out in SGBS preadipocytes, given the limitations imposed by the low number of primary cells that can be obtained from bariatric surgery, since this procedure is performed in our hospital via laparoscopy. In addition, as for other cell types, the efficiency of transfection for expression manipulation is higher in cell lines than in primary cells (especially in preadipocytes from IR/T2D obese individuals; new Figure 4A). We also restricted our studies to adipocyte differentiation given that this was the process that we found dysregulated after *PRPF8*/PRP8 overexpression and/or silencing. We sincerely hope that the Reviewing Editor and the Reviewers may understand our decisions.

Specifically, since SC preadipocytes from IR/T2D obese individuals showed diminished *SF3B1 tv1* expression levels (Figure 2—Figure supplements 1 and 2), we exposed SGBS cells to pladienolide-B, which binds to the SF3B complex and inhibits pre-mRNA splicing via targeting splicing factor SF3B1 (Cretu *et al.*, 2018, doi: https://doi.org/10.1016/j.molcel.2018.03.011; Aouida *et al.*, 2016, doi: 10.1016/j.biopen.2016.02.001). Notably, we found that exposure to pladienolide-B also impaired lipid accumulation in SGBS cells (new Figure 3—Figure supplement 3A). This concurred with, among other changes, diminished expression levels of *PPARG-1*, *PPARG-2*, and both total *SREBF1* and *SREBF1-1*, while *PPARG-4* and *BSCL2-2/3* expression increased (new Figure 3—Figure supplements 3B and 3C).

These findings support the involvement of the SF3B complex/SF3B1 in adipocyte differentiation and, together with our previous data on *PRPF8*/PRP8, suggest a prominent role for the major spliceosome in adipogenesis, which is consistent with the observation that this complex is responsible for splicing 99.5% of all introns in the human genome (Turunen *et al.*, 2013; doi: 10.1002/wrna.1141). Notably, although both *PRPF8*/PRP8 silencing and pladienolide-B treatment impaired lipid accumulation in differentiating SGBS adipocytes, their effects on gene expression levels were similar but not identical (i.e., *PPARG-4*, *BSCL^-^2*, *CIDEB-1*, *CIDEC-5*) (new Figures 4C-4G and Figure 3—Figure supplements 3B and 3C), indicating that the relative contribution of each component of the spliceosome core to adipocyte differentiation may differ, at least partially. This would be in accordance with previous evidence showing that depleting the cellular abundance of individual spliceosomal proteins modulates individual splice site usage (Dvinge *et al.*, 2019; doi: 10.1101/gr.246678.118).

To further explore the relative contribution of the components of the splicing machinery to adipocyte differentiation, we down-regulated the protein content of the splicing factor, SFPQ, by siRNA treatment of SGBS cells; rescue experiments using an expression vector coding for *SFPQ* were also carried out. As shown in new Figure 3—Figure supplements 3D-3G, either down-regulation or overexpression of *SFPQ* caused significant changes in the expression of only a few genes, and the combination of both strategies tended to restore the expression values found in control cells. To be more specific, *SFPQ* overexpression slightly reduced *CIDEB1* expression, while *SFPQ* silencing increased total *SREBF1* and *SREBF1-1* and decreased *PPARG-4* (new Figure 3—Figure supplements 3F-3G). Surprisingly, silenced SGBS cells exhibited reduced lipid content, as measured by the total number of lipid droplets per cell, mostly due to the increase in the number of lipid droplets (2.4-fold), while lipid droplet size decreased as compared with control cells (2.7-fold) (new Figure 3—Figure supplement 3E). A recent publication has reported “*fatty acid biosynthetic process*” as one of the Gene Ontology (GO) terms of the genes that are up-regulated upon loss of *SFPQ* expression (Gordon *et al.*, 2021; https://doi.org/10.1038/s41467-021-22098-z). This could be associated with the up-regulation of *SREBF1* in SGBS cells silenced for *SFPQ*, given the involvement of SREBP1 in adipocyte lipogenesis through its stimulatory effects on fatty acid production (Crewe *et al.*, 2019; doi: 10.1172/jci.insight.129397). The lack or small effect of *SFPQ* silencing on other genes relevant for adipocyte differentiation and/or lipid droplet biogenesis and growth may account for the increase in small lipid droplets and the decrease in total lipid content observed in *SFPQ*-silenced SGBS cells.

These new experiments and results have been described and discussed where appropriate in the new version of the manuscript (page 12, and page 22, respectively).

We sincerely hope that the new experiments that have been carried out for the revised version of the manuscript in both primary and SGBS preadipocytes, together with our previous data, will help convince the Reviewing Editor and the Reviewers that decreased *PRPF8*/PRP8 expression, as occurs in SC preadipocytes of IR/T2D *vs.* NG obese subjects, has specific consequences on adipocyte differentiation, likely by altering the function of the major spliceosome. We thank the Reviewers and the Reviewing Editor for their comments, which have been very helpful to reinforce our original proposal on the relevance of *PRPF8*/PRP8 and, for that matter, of the major spliceosome, as well as to show the contribution of other splicing components, such as SF3B complex and SFPQ, to adipocyte differentiation.

2 – While the study focused on comparing the molecular alterations between obese NG and obese IR/T2D, the levels of the genes/proteins of interest in lean subjects need to be considered. In particular, Figure 2E showed the expression levels of PRPF8 in lean subjects were significantly lower than that in the NG obese people. This would impose a critical caveat for the siRNA silencing studies because reduced PRPF8 level could represent the lean SC preadipocyte. Please discuss the overall implications of the findings when data from lean subjects are considered.

We concur with the Reviewing Editor that siRNA studies to reduce *PRPF8* levels may seem paradoxical in the face of the reduced *PRPF8* mRNA content observed in lean subjects as compared to that found in the NG obese people. As shown in the original Figure 2 and the new Figure supplement 6 that has been prepared to address Minor Comment 4, not only *PRPF8* but also other component of the major spliceosome, *SF3B1 tv1*, and the splicing factors, *CELF1* and *SNW1*, showed significantly lower expression levels in lean SC preadipocytes than in their counterparts from obese individuals of cohort 2 (i.e., individuals with simple obesity, BMI 30-35 kg/m^2^). These results indicate that, as compared with lean individuals, NG simple obesity involves the up-regulation of splicing-related genes in SC preadipocytes, which might be required to sustain the increase in adipocytes that has been reported to be associated with healthy adipose tissue expansion (reviewed by Vishvanath and Gupta, 2019; https://doi.org/10.1172/JCI129191), given the role of splicing in shaping cell identity (Dvinge, 2018; https://doi.org/10.1002/1873-3468.13119).

Interestingly, our original studies on *PRPF8*/PRP8 and other splicing-related genes together with our new data demonstrate that, either in simple obesity (Figure 2C, and new Figure 2—Figure supplement 3; cohort 2) or morbid obesity (Figure 2B, and Figure 2—Figure supplements 1 and 2; cohort 1), IR/T2D is accompanied by diminished expression levels of these genes as compared to NG obesity. Given that both simple and morbidly obese NG and IR/T2D groups were matched by BMI, these findings suggest that the adaptive response of SC preadipocytes to adipose tissue expansion (i.e., increased expression of splicing-related genes), is lost in the transition from NG to IR/T2D in both obese and extremely obese individuals. There is only one exception to this rule, the minor spliceosome component, *RNU12*, whose levels were higher in IR/T2D than in NG obesity (simple or morbid) (original Figure supplements 4 and 5, cohort 1; and new Figure 2—Figure supplement 3B, cohort 2). These results might be related to the differences existing between the minor and major spliceosome (Akinyi and Frielander, 2021; doi: 10.3389/fgene.2021.700744), but since the former is less known than the latter and there is not much information on the specific role played by *RNU12*, in particular in adipocytes, we modestly believe that there is no evidence enough to discuss these findings.

The new results have been depicted in the revised version of the manuscript (pages 8 and 9, and page 19 and 20, respectively). In all, our findings support the notion that SC preadipocyte dysfunction and IR/T2D occur when the expression levels of important components of the splicing machinery remain below (or, occasionally, above) those needed to respond to the requirements imposed by the obesogenic environment. With all due respect, we modestly believe that, in this scenario, our silencing studies of *PRPF8* are justified because they mimic the reductions in gene expression observed under conditions of IR/T2D *vs*. NG in both simple and severe obesity. Our novel data on *SFPQ* silencing and SF3B inactivation further support the notion that decreasing the splicing potential of adipocyte precursors impairs adipogenesis.

3 – The authors report increased expression of components of the ERAD pathway in preadipocytes from obese TD2 patients. While this doesn't seem to be associated with a consistent increase in UPR response, the authors conclude that in cells obtained from this subgroup of patients there is an impairment in proteostasis. Further mechanistic studies have been performed in a completely independent model (SGBS cell lines) but have not been replicated in preadipocytes obtained from patients (that are the subject of this study). To validate the contribution of ERAD and proteolysis to the functional impairment of preadipocytes in obese patients, these experiments should be performed in preadipocytes obtained from patients. Is proteasome activity decreased in preadipocytes from obese TD2 patients? Is it normalized by chemical chaperones? Is proliferation and/or differentiation potential of SC preadipocytes from obese TD2 patients recovered by chemical chaperones? Does Bip play a central role in orchestrating this response in these cells?

We thank the Reviewing Editor and the Reviewers for their insightful recommendations. As for the splicing machinery, we performed novel experiments in primary preadipocytes from NG, IR, and T2D obese individuals, in this case preadipocytes isolated from omental fat, to validate the contribution of ERAD and proteostasis to the functional impairment of preadipocytes in obese individuals. Regarding OM preadipocytes from NG obese individuals, we took advantage of the knowledge gained from SGBS cells in the original version of the manuscript (original Figures 5I-5K) and employed three experimental groups: (1) exposure to HGHI conditions, (2) to TUDCA, or (3) to the combination of both (new Figure 7). We observed that, as for SGBS cells, exposure of OM preadipocytes from NG obese individuals to HGHI increased the expression levels of genes involved in all ERAD steps (recognition, retrotranslocation, ubiquitination, and targeting to the proteasome) in these cells (new Figures 7A-D). On the other hand, TUDCA, which did not alter essentially the expression of ERAD genes when administered alone, reduced HGHI-induced gene expression increases to control levels (i.e., OM preadipocytes from NG obese individuals exposed to medium alone) (new Figures 7A-D). As for SGBS cells, HGHI increased the accumulation of ubiquitinated proteins in OM preadipocytes from NG obese individuals, and only a slight, not significant decrease in ubiquitin-conjugated proteins was observed when HGHI was combined with TUDCA (new Figure 7E). Notably, the activity of the 26S proteasome remained unchanged in all the experimental conditions tested (new Figure 7F). Microscopic analysis of NG OM preadipocytes under the different experimental settings demonstrated an inhibitory effect of HGHI on lipid accumulation, that was reverted by TUDCA (new Figure 7G). In accordance with these results are also our new data on mRNA levels of adipogenic genes in NG OM preadipocytes exposed to HGHI alone or in combination with TUDCA (new Figures 7H and 7I).

In clear contrast, and as the Editor correctly anticipated, our new experiments demonstrate that, as compared to their NG counterparts, proteasome activity is decreased in OM preadipocytes from both IR and T2D obese individuals, who also showed increased levels of ubiquitin-conjugated proteins (new Figures 7E and 7F). Interestingly, TUDCA reverted both effects in OM preadipocytes from IR obese patients but not in T2D preadipocytes (new Figures 7E and 7F). TUDCA was also able to increase the expression of markers of differentiation in preadipocytes from IR and, to a lower extent, also from T2D obese individuals (new Figures 7H and 7I). In fact, this chaperone was able to increase the lipid content of OM preadipocytes of IR and T2D obese subjects to the levels found in OM preadipocytes of NG obese subjects, though TUDCA-treated T2D OM preadipocytes exhibited low LD numbers (new Figure 7G). Together, these results suggest that preadipocyte proteostasis is more severely compromised in OM fat of T2D than in IR obese individuals and that chemical chaperones may improve adipogenesis specially in the latter.

These new data have been depicted in the revised version of the manuscript and discussed where appropriate (pages 17-19, and pages 25 and 26).

Finally, due to the high number of cells employed for the experiments depicted above, and the limitations imposed by the number of IR and T2D patients available with biochemical and anthropometric parameters comparable to those employed for the original version of the manuscript, we were only able to carry out overexpression experiments of BiP in primary OM preadipocytes from NG obese patients. These samples were employed for measurement of ubiquitinated proteins as a readout of cell proteostasis. As shown in Author response image 1, BiP overexpression highly increased the amount of ubiquitinated proteins in NG preadipocytes, which is consistent with our previous results in SGBS cells showing that pCMV-BiP expression caused an imbalance in both ERAD and adipogenesis (original Figure supplement 8; new Figure 7—Figure supplement 3). In accordance with this notion are our results showing increased BiP levels in NG preadipocytes exposed to HGHI as well as in IR and T2D preadipocytes as compared to control NG preadipocytes (new Figure 7A), which is consistent with the increases observed in the expression of ERAD genes and ubiquitinated proteins in these groups (Figure 6). TUDCA did not prevent the response to pCMV-BiP transfection, which might be linked to the strong effect caused by BiP overexpression on the accumulation of ubiquitinated proteins (6-fold increase *vs*. mock transfected cells; Author response image 1).

**Author response image 1. sa2fig1:** Representative dot-blot and protein quantification of ubiquitin-conjugated proteins in OM preadipocytes from NG obese individuals in control conditions (control, white), overexpressing BiP (pCMV-BIP, black), exposed 14 h to 0. 5 mg/mL TUDCA (TUDCA, grey), or to a combination of both (TUDCA+pCMV-BIP, white with black dots) (6-7 replicate studies, 1 technical replicate each). ******P<0.01 *vs.* control; **^###^**P<0.001 *vs.* pCMV-BIP; **^$^**P<0.05 *vs.* TUDCA. Data are presented as mean ± standard error of the mean (S.E.M.), n = 6. Kruskal-Wallis with Dunn's multiple comparisons test was used. Normality distribution was determined by Shapiro-Wilk normality test.

Since the new results on BiP are not conclusive and essentially correlative, and because we could not check whether BiP silencing in IR/T2D preadipocytes could revert ERAD up-regulation, proteasome inactivation, and accumulation of ubiquitinated proteins in these cells, we cannot discuss at present on a potential central role of BiP in orchestrating the response of preadipocytes in IR/T2D obesity. Therefore, we have removed our original proposal on the potential role of BiP as a functional link between UPR and ERAD in preadipocytes (page 20, lines 468-470 in the original version) in the revised version of the manuscript.

We sincerely hope that our results on SGBS cells, together with the new data that have been generated using primary preadipocytes isolated from samples of obese individuals with NG, IR, and T2D, will help convince the Editor and Reviewers on the relevance of both alternative splicing and ERAD on preadipocyte function and dysregulation in obesity-associated metabolic disease.

4 – The rigor of data presentation: For the data presented, please show each individual value using the dot plot to let the readers see the data's distributions and ranges of variations. For in vitro studies, please indicate the numbers of technical replicates within a given experiment and the numbers of replicate studies. Please elaborate on how Western blots' data were quantified, including imaging quantification and normalization of loading.

As recommended, all the graphs including data from human primary cells have been modified so that individual values are now shown using dot plot. We have left the original graphs for representing time-series data (i.e., quantification of gene expression values during differentiation) to facilitate data visualization, except in the case of *PRPF8* mRNA values in SC and OM differentiating preadipocytes, which are included in a main figure (revised Figure 2D).

Following the recommendation offered by the Editor, we have indicated the numbers of technical replicates within each experiment and the numbers of replicate studies for the in vitro studies in the Figure Legends. The number of replicate studies employed for the in vitro experiments was mentioned in the Statistical Analysis Section (page 42, lines 1008-1009 in the original version of the manuscript). We apologize for not having mentioned this information or included the number of technical replicates in the original figure legends.

Regarding Western blots' data quantification, we followed a protocol employed by us in previous studies that have been published in prestigious journals in the field of proteomics (Peinado *et al.*, 2011; Mol. Cell Proteomics, doi: 10.1074/mcp.M111.008094) or metabolism (Jimenez-Gomez *et al.*, 2013, Cell Metab, doi: 10.1016/j.cmet.2013.09.004), that is, all the experiments were carried out loading equal amounts of proteins from all the experimental groups and values were further normalized against Ponceau staining. In this regard, we and other authors have shown that both actin and other common reference standards (tubulin, GAPDH) exhibit significant intra-group variations in proteomic studies of adipose tissue (Peinado *et al.*, 2010, doi: 10.1002/pmic.201000350; Diaz-Ruiz *et al.*, 2014, doi: 10.1089/ars.2014.5939; Guzman-Ruiz *et al.*, 2020, doi: 10.1096/fj.201902703R), which prevented us from using these markers as standard proteins for further quantitative immunoblotting analyses. Indeed, immunoblotting using antibodies against b-actin, tubulin, or GADPH does not always give consistent results when using protein extracts from adipose tissue. We apologize if the information provided was not clear enough for understanding the procedure employed for Western blots' data quantification. We have rephrased the paragraph in the revised version of the manuscript to indicate that quantification of band intensities was carried out on digital images of membrane samples provided by LAS4000 gel documentation system (GE Healthcare; Barcelona, Spain) using ImageJ (section “*Quantitative immunoblotting*” in Material and Methods**,** page 45 in the revised version of the manuscript).

5 – The authors report two mechanisms linked to dysfunction of adipocyte differentiation and function in obese subjects with IR or T2D (i.e., the spliceosome and the ERAD). Can the authors discuss and clarify whether each of these pathways plays an essential, complementary, or redundant role in the pathologies of adipocytes dysfunction in IR or T2D subjects? It would be informative to include an integrative model to present the two pathways and how they regulate adipocyte differentiation and function.

Following the recommendation offered, we have replaced the schematic representation of the proposed mechanism of action of PRP8 on adipocyte differentiation from Figure 3J by a new Figure 8 wherein the two pathological pathways described to be altered in preadipocytes from IR/T2D obese individuals are depicted. We sincerely hope that this new integrative model may be helpful to further understand our results on how impaired splicing and ERAD/ER proteostasis may prevent adipocyte differentiation.

According to our data and as depicted in our responses to Essential Revisions 1 and 3, alteration of either of these two processes in primary preadipocytes induces cell dysfunction and impaired adipogenesis (new Figures 4 and 7). This supports the notion that, being both processes essential, at least in SC preadipocytes, they may play redundant roles in the pathologies of adipocytes dysfunction in IR or T2D subjects (in terms of impaired adipogenesis). However, these two pathways may not be intermingled since, according to new studies that have been carried out to address the comments offered by the Editor, *PRPF8* down-regulation by siRNA treatment did not alter the expression of ERAD genes in preadipocytes. These results are included in the Author response image 2.

**Author response image 2. sa2fig2:** mRNA levels of ERAD-related genes in SGBS preadipocytes three post-transfection (differentiation day 7) with control (white) or PRPF8-siRNA (black) (6 replicate studies, 4 technical replicates each). ******P<0.01 *vs.* control-siRNA. Data are presented as mean ± standard error of the mean (S.E.M.). Unpaired t test or Mann Whitney test (for parametric or non-parametric data, respectively) were used. Normality distribution was determined by Shapiro-Wilk normality test.

In turn, in our original manuscript we showed that all the obesogenic IR insults tested in our study, including exposure to HGHI, TNFa, or palmitate, up-regulated the expression of ERAD genes in SGBS cells (original Figure 5). We have tested the effects of these treatments on *PRPF8* mRNA levels and observed that none of these experimental settings had any effect on *PRPF8* expression . In all, these results indicate that splicing and ERAD are differentially regulated by external stimuli. We have referred to the results concerning *PRPF8* expression in the in vitro models as data not shown (page 23, in the revised version of the manuscript).

We cannot exclude the possibility that *PRPF8*/PRP8 impairment may be the result of the action of other pathogenic processes causing adipose tissue dysfunction in obese individuals. We have included fibrosis, as well as other hallmark features of the IR obese adipose tissue (i.e., hypoxia) as potential drivers of splicing dysregulation in preadipocytes in the new Figure 8 that has been prepared for the revised version of the manuscript.

With all due respect, we believe that, in all, our data support the accepted notion that (pre)adipocyte dysfunction in obesity may be triggered by distinct environmental insults (pro-inflammatory cytokines, lipids, glucose, hypoxia, fibrosis) and intracellular signals (increased lipid content, mechanotransduction pathways) which, in turn, may activate either common or different cellular stress processes (ER stress, altered UPR/ERAD, proteasome malfunction, splicing, oxidative stress, inflammatory response, autophagy, etc) to trigger pathogenic responses in adipose tissue cells

**Author response image 3. sa2fig3:** mRNA levels of Prpf8 in differentiated 3T3-L1 adipocytes (day 10 of differentiation) cultured in 3D matrices based on collagen I-enriched hydrogels containing or not increasing concentrations of the proteoglycan, lumican (0-30 ng/mL). *P<0.05 vs. 0 ng/mL lumican. Data are presented as mean ± standard error of the mean (S.E.M.), n = 3. One-way ANOVA with Tukey’s multiple comparisons test was used. Normality distribution was determined by Shapiro-Wilk normality test.